# Replication study on the role of dopamine-dependent prefrontal reactivations in human extinction memory retrieval

Elena Andres [1,8] ✉, Hu Chuan-Peng [1,2,8], Anna M. V. Gerlicher[3,4,5], Benjamin Meyer[1,3], Oliver Tüscher [1,6,7] & Raffael Kalisch [1,3]

Even after successful extinction, conditioned fear can return. Strengthening the consolidation of the fear-inhibitory safety memory formed during extinction is one way to counteract return of fear. In a previous study, we found that post-extinction L-DOPA administration improved extinction memory retrieval 24 h later. Furthermore, spontaneous post-extinction reactivations of a neural activation pattern evoked in the ventromedial prefrontal cortex (vmPFC) during extinction predicted extinction memory retrieval, L-DOPA increased the number of these reactivations, and this mediated the effect of L-DOPA on extinction memory retrieval. Here, we conducted a pre-registered replication study of this work in healthy male participants. We confirm that spontaneous post-extinction vmPFC reactivations predict extinction memory retrieval. This predictive effect, however, was only observed 90 min after extinction, and was not statistically significant at 45 min as in the discovery study. In contrast to our previous study, we find no evidence that L-DOPA administration significantly enhances retrieval and that this is mediated by enhancement of the number of vmPFC reactivations. However, additional non-preregistered analyses reveal a beneficial effect of L-DOPA on extinction retrieval when controlling for the trait-like stable baseline levels of salivary alpha-amylase enzymatic activity. Further, trait salivary alpha-amylase negatively predicts retrieval, and this effect is reduced by L-DOPA treatment. Importantly, the latter findings result from non-preregistered analyses and thus further investigation is needed.

Recognizing an event or situation as dangerous and reacting defensively when it re-occurs is a learning mechanism (fear or threat conditioning) that is fundamental for survival. However, learning when a previous threat no longer signals danger and ceasing costly defense behavior (fear or threat extinction) also has great adaptive value and has been related to a reduced risk for threat-related mental disorders such as post-traumatic stress disorder (PTSD) or anxiety disorders[1]. Also, extinction learning is the probable mechanism that

[1]Leibniz Institute for Resilience Research (LIR), 55122 Mainz, Germany. [2]School of Psychology, Nanjing Normal University, 210024 Nanjing, China. [3]Neuroimaging Center (NIC), Focus Program Translational Neuroscience (FTN), Johannes Gutenberg University Medical Center Mainz, 55131 Mainz, Germany. [4]Department of Experimental Psychology, Helmholtz Institute, Utrecht University, 3584 CS Utrecht, the Netherlands. [5]Department of Clinical Psychology, University of Amsterdam, 1018 WS Amsterdam, the Netherlands. [6]Department of Psychiatry and Psychotherapy, Johannes Gutenberg University Medical Center, 55131 Mainz, Germany. [7]Institute of Molecular Biology (IMB), 55128 Mainz, Germany. [8]These authors contributed equally: Elena Andres, Hu Chuan-Peng. ✉e-mail: elena.andres@lir-mainz.de

underlies the exposure-based treatment of threat-related disorders[2,3].

To investigate whether fear extinction lastingly reduces conditioned responses (CRs), a three-phase paradigm is commonly used, consisting of fear conditioning, fear extinction, and a memory test. During fear conditioning, an innocuous conditioned stimulus (CS) is repeatedly paired with an unconditioned stimulus (US). Participants start to exhibit CRs to the formerly neutral CS, and a CS-US association, or 'fear memory', is formed[4]. During extinction, participants are re-exposed to the CS in the absence of the US several times, and CRs decrease. During the test phase, participants are once more exposed to the CS, again in the absence of the US, and CRs are measured. Based on this paradigm, studies have reported that even after a complete reduction of CRs over the course of extinction, CRs often return during the test phase ("return of fear"[5]). Thus, extinction learning is not an unlearning or erasure of the original fear memory, but formation of a new CS-noUS association or 'extinction memory'[5], and the test phase effectively examines the retrieval and/or expression of the fear in competition with the extinction memory.

In exposure-based treatment, relapse after successful exposure is not uncommon, and return of fear is presumably a precursor for relapse[3]. Therefore, it would be highly desirable to develop methods to prevent the return of fear. Given that consolidation processes are crucial for long-term memory expression[6–8], reinforcing the consolidation of the extinction memory may be one promising avenue[9].

The dopaminergic system plays a crucial role in memory consolidation[10]. In the case of extinction, memory retrieval has been shown to be worse when dopaminergic activity is decreased after extinction training. After microinfusion of a D1 receptor antagonist into the infra-limbic part of the medial prefrontal cortex (IL) following extinction training in rats, later extinction memory retrieval was reduced relatively to a vehicle condition, suggesting impaired consolidation[11]. Conversely, increasing dopaminergic activity after extinction training results in better long-term extinction memory retrieval. So, the systemic administration in mice of methylphenidate, a dopamine and noradrenaline reuptake inhibitor, after extinction learning led to relatively decreased fear responses at test 24 h later[12]. Of particular relevance for potential clinical applications, post-extinction systemic administration of the anti-Parkinson drug L-DOPA, a precursor of dopamine that preferentially enhances dopaminergic turnover in the frontal cortex[13], improved extinction memory retrieval both in mice and healthy normal humans in altogether six experiments[14–16]. A negative result in one human study was accompanied by reduced neural activity at test in brain areas related to conditioned fear[17]. In two human experiments conducted outside the magnetic resonance imaging (MRI) environment[18], post-extinction L-DOPA reduced CRs at test only in participants who showed successful extinction learning; in one other non-MRI study, the L-DOPA effect was not significant[19]. Hence, L-DOPA is a likely pro-consolidation agent, although the boundary conditions for its effectiveness still have to be established.

A strong body of research supports a causal role for the IL, the rodent homolog of the vmPFC, in the consolidation of extinction memories[20]. In humans using functional MRI (fMRI), Gerlicher et al.[16] reported that a neural activation pattern in the vmPFC, which they initially observed during extinction when the US was unexpectedly omitted from CS presentations, spontaneously re-occurred during post-extinction rest. They further observed that the number of reactivations of this multi-voxel pattern (MVP) in the resting state predicted extinction memory retrieval as well as vmPFC activation at the memory test 24 h later. Importantly, the IL/vmPFC is a target of dopaminergic projections from the ventral tegmental area (VTA), and extinction has been shown to evoke lasting dopamine release in this brain area[21]. Extending the rodent microinfusion data[11], Gerlicher et al.[16] also reported that post-extinction L-DOPA administration

enhanced the number of spontaneous vmPFC MVP reactivations and that this mediated the beneficial effect of L-DOPA on extinction retrieval. No other brain area showed a statistically significant relationship, suggesting a pro-consolidation action of dopamine that depends on consolidation-related vmPFC activity.

A further intriguing aspect of this study was that all findings were specific to vmPFC MVPs from early extinction trials. Early in extinction, the CS still elicits a high US expectation, and the surprise (or "prediction error", PE) generated by US omission is highest[22]. It is generally accepted that the formation of the new CS-noUS association in extinction is driven by this PE[23–25], and extinction PEs in turn have been shown to be encoded by phasic dopamine release in the ventral striatum[26]. It is currently unclear whether the vmPFC also receives dopaminergic extinction PE signals, but extinction PE-correlated fMRI activity has been observed in humans also outside the ventral striatum[27]. Taken together, the VTA-originating dopaminergic system may tie together extinction learning, memory formation, and memory consolidation via influences on the prefrontal cortex.

We here aimed to directly replicate the findings of Gerlicher et al.[16], as preregistered in Chuan-Peng et al.[28]. For this, we employed the same fMRI paradigm (Fig. 1a) with differential fear conditioning in context A (background picture) on day 1. Two geometric symbols served as CS+ (reinforced at its offset in 50% of conditioning trials with a painful US) and CS- (non-reinforced CS), respectively. Extinction learning in context B on day 2 was immediately followed by oral placebo or L-DOPA administration (randomized between-subject design) and resting-state fMRI scans 10, 45, and 90 min after the end of extinction, to cover the early consolidation window. On day 3, we tested extinction memory retrieval in context B. During extinction and test, no US was administered. Skin conductance responses (SCRs) were used as CRs (Fig. 1b).

We tested three main replication hypotheses[28]: First, we expected that post-extinction L-DOPA as compared to placebo administration would improve extinction memory retrieval at test on day 3. That is, we expected smaller differential (CS + > CS−) SCRs for L-DOPA- compared to placebo-treated participants (hypothesis 1). Second, we expected that the number of spontaneous post-extinction reactivations during the post-extinction resting-state scans of a vmPFC MVP linked with CS+ offsets in the first five extinction trials (first third of extinction) would predict extinction memory retrieval during test on day 3 in both placebo- and L-DOPA-treated participants (hypothesis 2). We were particularly interested in the resting-state scan conducted 45 min after extinction, as this is where Gerlicher et al.[16] had observed a predictive relationship to retrieval, but also allowed ourselves to test the scans conducted 10 and 90 min after extinction, as a precise timing of, or a very restricted time window for, the effect was considered unlikely. Third, we expected that L-DOPA would significantly increase the number of vmPFC reactivations (hypothesis 3).

To determine the optimal replication sample size, we combined different approaches[28]. We conducted a power analysis based on the critical effect size from Gerlicher et al.[16] for the most important hypothesis 1 (see Methods for details). This led to a required sample size of $N = 22$ (i.e., eleven participants per treatment group). However, considering that effects in discovery samples may overestimate the true effect size, Simonsohn[29] suggested that replication studies should have a sample size that is at least 2.5 times greater than that of the discovery study ($N = 40$ in Gerlicher et al.[16]), leading to a required sample size of $N = 100$ (50 per group). This approach would mean that we would need a sample size 4.5 times bigger than the sample size estimated based on the critical effect size in the discovery study, which we considered exaggerated. To balance the feasibility of a 3-day pharmacological fMRI paradigm and the requirement to limit the number of participants exposed to study-associated burden[30] with the requirement of sufficient statistical power, we settled on a sample size of $N = 70$ (35 per group). With an estimated drop-out rate of 10%, this

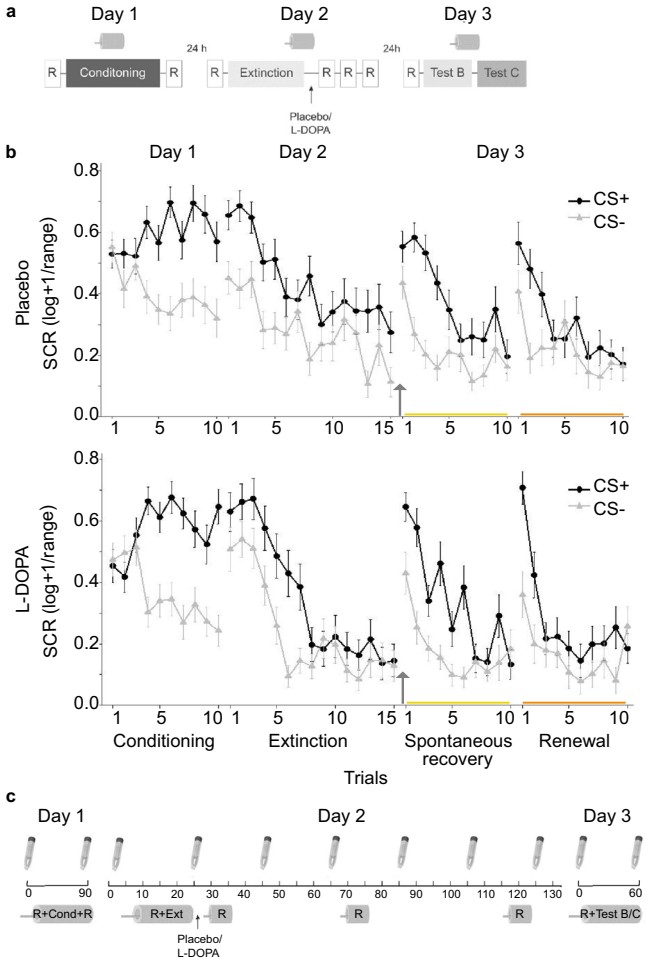

**Fig. 1 | Experimental design and skin conductance responses. a** A 3-day fMRI paradigm was employed with fear conditioning on day 1 in context A, extinction on day 2 in context B, and test on day 3 first in context B (the context in which the extinction learning took place), and then in a new context C. Immediately after extinction learning, participants were administered either a placebo or a L-DOPA pill. Drug administration was randomized and double-blinded (placebo: *n* = 35, L-DOPA *n* = 35, all male, for group characteristics, see Supplementary Table 1). All three phases occurred in the MRI scanner. Further, resting-state scans (R) were collected before and after fear learning, before the start of extinction, -10, 45 and 90 min after extinction, and before test. **b** Conditioned responses (CRs) were assessed as SC responses (SCRs) to the CS+ and CS− throughout all experimental phases. The arrow depicts pill administration. The upper panel shows the mean SCRs of the placebo group (*n* = 33/27/30 on days 1/2/3), the lower panel shows results of the L-DOPA-treated group (*n* = 33/24/25 on days 1/2/3). Data is presented as mean ± standard error of the mean (sem). **c** Saliva samples were collected on day 2 before the start of the experimental paradigm (time 0 min) followed by a resting-state scan (approx. 8 min) and extinction (approx. 15 min). Subsequent saliva samples were taken right after extinction before pill intake and then about every 20 minutes (+/− 2 min), with the last sample taken after the last resting-state measurement at 90 min post-extinction. Samples on day 1 and 3 were collected before and after the paradigm in the scanner. Tubes represent saliva sample collections. Scanner bore symbols represent scanning times. Cond Conditioning, Ext Extinction.

means we could expect to achieve a final sample size of *N* = 63, which is 1.5 times greater than in the discovery study and nearly 3 times greater than based on power calculation. The responsible medical ethics committee approved this approach.

We furthermore extended the paradigm to clarify two remaining questions (see preregistration[28]). First, to assess whether L-DOPA not only protects against the spontaneous recovery of extinguished fear

after a mere passage of time but also against the renewal of fear gated by a context change[5], we added a test in a new context C (non-extinction context, a new background picture) on day 3 (secondary research question 1; Fig. 1a). Protection against renewal would suggest that L-DOPA makes the extinction memory context independent, as indicated by the results of Haaker et al[14]. Second, we collected saliva samples throughout all experimental days to measure enzymatic activity levels of salivary alpha-amylase (sAA) and the concentration of salivary cortisol (sCORT) as markers of the activity of the sympathetic nervous system (SNS) and of the hypothalamus-pituitary-adrenal gland (HPA) axis, respectively[31,32] (Fig. 1c). The activity of these systems, linked to arousal and stress, have been suggested to affect consolidation processes in general and extinction consolidation in particular[33–35]. We had previously observed L-DOPA main effects in two out of three MRI experiments[14,16,17], while two purely behavioral experiments showed only conditional L-DOPA effects, moderated by prior extinction success[18], and one other behavioral study showed no significant effect[19]. This might indicate a potential influence of the MRI environment on the effect of L-DOPA[19]. Previous studies have reported relatively increased sAA[36,37] and sCORT levels[36,38] in the scanner environment. It is conceivable that higher arousal or stress might impair extinction[33,39], leaving more room for an augmenting effect of L-DOPA; alternatively, dopamine might positively interact with arousal- or stress-related neural activity in its effect on extinction consolidation[13,33]. We, therefore, wondered if peak sAA or sCORT levels on the extinction day 2 would affect extinction retrieval on day 3 and interact with treatment (placebo vs. L-DOPA) (secondary research question 2).

sAA and sCORT have been observed to increase in response to arousing and stressful stimuli, such as being scanned[36,38,40] or receiving painful stimulation[41–43], but they have also been reported to exhibit pronounced stability across time when repeatedly measured in baseline states of rest[44–46]. For sAA in particular, variance across a day or across days is better explained by between- than within-subject differences[44,47]. Out et al. have also demonstrated that baseline sAA is substantially heritable[48]. Hence, repeated baseline sAA assessments (as well as repeated sCORT measures) may also be employed as trait-like individual-differences markers (trait sAA, trait sCORT). These individual differences may partly determine individual differences in extinction and/or in the effect of L-DOPA on extinction memory consolidation. At the request of one reviewer, we therefore also tested whether sAA and sCORT baseline levels on the three experimental days interact with extinction learning, extinction retrieval, and drug treatment (non-preregistered secondary research question 3).

In this work, we show that the number of spontaneous reactivations in a resting-state scan 90 min after extinction of an MVP elicited specifically in the vmPFC during early extinction learning when the US is omitted at CS+ offsets positively predicts extinction memory retrieval 24 h later. We also find that trait sAA as well as peak sAA at the time of extinction negatively predict retrieval. Controlling for trait and state sAA reveals a beneficial L-DOPA main effect and a trait sAA by group interaction, such that the negative effect of trait sAA is rescued by L-DOPA treatment. Our findings suggest that boosting dopaminergic activity promotes the consolidation of extinction in individuals with elevated basal SNS activity.

## Results
### Effect of L-DOPA administration post extinction on extinction memory retrieval
First, we tested whether post-extinction L-DOPA compared to placebo administration on day 2 enhances extinction memory retrieval on day 3 (preregistered hypothesis 1). We compared the extent of spontaneous recovery of CRs in the extinction context B between the two treatment groups (see Fig. 1b). In addition, we asked whether L-DOPA administration also reduces contextual renewal of fear in a

new context C on the test day (preregistered secondary research question 1).

Fear acquisition on day 1 was equally successful in both groups, as indicated by a significant effect of stimulus (CS + > CS- SCRs averaged across last 20% trials: $F_{1,64} = 70.52$, $p < 0.001$, generalized $\eta^2 = 0.23$) in the absence of significant group (placebo/L-DOPA) and stimulus by group effects (group: $F_{1,64} = 0.96$, $p = 0.329$; stimulus*group: $F_{1,64} = 0.58$, p = 0.449, $n = 66$ participants with sufficient SCR data quality; Fig. 1b; see Methods and Supplementary Table 2 for details on exclusions and resulting final sample sizes per analysis). Fear was retrieved at the beginning of extinction on day 2 in both groups (start-fear: averaged CRs across first 20% trials; stimulus: $F_{1,49} = 36.23$, $p < 0.001$, generalized $\eta^2 = 0.16$; group: $F_{1,49} = 0.48$, $p = 0.493$; stimulus*group: $F_{1,49} = 2.30$, $p = 0.136$; $n = 51$). By the end of extinction on day 2 (end-fear: SCRs across last 20% trials), an unforeseen group difference emerged (stimulus: $F_{1,49} = 8.81$, $p = 0.005$, generalized $\eta^2 = 0.04$; group: $F_{1,49} = 2.07$, $p = 0.157$; stimulus*group: $F_{1,49} = 4.92$, $p = 0.031$, generalized $\eta^2 = 0.02$, $n = 51$), characterized by higher CRs towards the CS+ in the placebo group (two-sample $t$-test: $t_{49} = 2.09$, $p = 0.041$, CI 95% [0.006 0.31]).

Contrary to our predictions, administration of L-DOPA on day 2 did not result in a statistically significant reduction of spontaneous recovery on day 3 (averaged CRs across all trials; stimulus: $F_{1,53} = 55.30$, $p < 0.001$, generalized $\eta^2 = 0.20$; group: $F_{1,53} = 0.91$, $p = 0.344$; stimulus*group: $F_{1,53} = 0.02$, $p = 0.887$; $n = 55$). Bayesian analysis indicated weak evidence in favor of H0 ($BF_{01} = 3.85$). Also, contextual renewal of fear was not significantly reduced (stimulus: $F_{1,53} = 23.66$, $p < 0.001$, generalized $\eta^2 = 0.07$; group: $F_{1,53} = 0.91$, $p = 0.345$, stimulus*group: $F_{1,53} = 0.17$, $p = 0.685$, $BF_{01} = 3.44$). Adjusting for the observed group differences at the end of extinction did not change the result (multiple regression including end-fear at extinction as well as fear acquisition and start-fear at extinction as covariates, non-preregistered analysis: spontaneous recovery: $\beta_{group} = 0.02$, SE = 0.05, $t_{40} = 0.51$, $p = 0.615$; renewal: $\beta_{group} = 0.02$, SE = 0.04, $t_{40} = 0.55$, $p = 0.583$; $n = 45$), although it showed in both cases that end-fear at extinction significantly and positively predicted CRs at test (spontaneous recovery: $\beta_{end-fear} = 0.25$, SE = 0.10, $t_{40} = 2.52$, $p = 0.016$; renewal: $\beta_{end-fear} = 0.27$, SE = 0.09, $t_{40} = 3.06$, $p = 0.004$; Fig. 2), as also observed in ref. 16.

Since preregistration, three studies have reported that better extinction (lesser end-fear) is associated with better extinction retrieval in L-DOPA-treated, but not placebo-treated, participants, suggesting the L-DOPA effect may under some circumstances be restricted to successful extinguishers[18,19]. However, in an additional non-preregistered interaction analysis, group did not significantly interact with end-fear of extinction in predicting CRs at spontaneous recovery ($\beta_{group*end-fear} = 0.01$, SE = 0.20, $t_{41} = 0.07$, $p = 0.948$) or renewal ($\beta_{group*end-fear} = -0.15$, SE = 0.18, $t_{41} = 0.85$, $p = 0.400$). Instead, comparable relationships between end-fear at extinction and CRs at the spontaneous recovery and renewal tests were found in both groups (simple slope analysis: spontaneous recovery: $\beta_{end-fear/placebo} = 0.24$, SE = 0.13, $t_{39} = 1.86$, $p = 0.070$; $\beta_{end-fear/L-DOPA} = 0.26$, SE = 0.16, $t_{39} = 1.65$, $p = 0.110$; renewal: $\beta_{end-fear/placebo} = 0.32$, SE = 0.11, $t_{39} = 2.82$, $p = 0.008$; $\beta_{endf-ear/L-DOPA} = 0.19$, SE = 0.14, $t_{39} = 1.35$, $p = 0.185$).

Together, these findings do not confirm an effect of L-DOPA on extinction memory consolidation but show a relationship between extinction success and extinction memory retrieval.

## Relationship between vmPFC pattern reactivations post extinction and extinction memory retrieval

Next, we tested whether spontaneous post-extinction reactivations of a multi-voxel activity pattern (MVP) in the vmPFC, elicited by the offset of the first five CS+ trials early in extinction, on day 2 predict extinction memory retrieval at test on day 3 (preregistered hypothesis 2). We observed the predicted negative correlation between the number of vmPFC reactivations and CRs in the spontaneous recovery test for the resting-state scan conducted 90 min after extinction ($\beta = -0.10$, SE = 0.03, $t_{41} = 3.17$, $p = 0.003$, Bonferroni threshold for testing four time points: 0.013, $n = 46$; Fig. 3), but not for the scans conducted before as well as 10 and 45 min after extinction (before: $\beta = -0.08$, SE = 0.07, $t_{41} = 1.13$, $p = 0.267$; 10 min: $\beta = 0.02$, SE = 0.04, $t_{41} = 0.61$, $p = 0.548$; 45 min: $\beta = 0.03$, SE = 0.03, $t_{41} = 0.84$, $p = 0.407$). Like in the discovery study, where the effect was observed at 45 min, the relationship did not differ significantly between groups (interaction number of reactivations at 90 min*group: $p = 0.659$, simple slope analysis: $\beta_{placebo} = -0.09$, SE = 0.04, $t_{39} = 2.13$, $p = 0.039$, $\beta_{L-DOPA} = -0.12$, SE = 0.05, $t_{39} = 2.36$, $p = 0.023$). The relationship did not extend to renewal (non-preregistered analysis, before: $\beta = -0.07$, SE = 0.06, $t_{41} = 1.22$, $p = 0.229$; 10 min: $\beta = -0.002$, SE = 0.03, $t_{41} = 0.05$, $p = 0.965$; 45 min: $\beta = 0.008$, SE = 0.03, $t_{41} = 0.29$, $p = 0.773$, 90 min: $\beta = -0.004$, SE = 0.03, $t_{41} = 0.13$, $p = 0.899$).

Additional analyses confirmed the anatomical specificity of the relationship at 90 min to the vmPFC (Supplementary Fig. 2a–h). Also at 45 min, no region emerged as showing a statistically significant relationship (Supplementary Fig. 3a–i). The relationship at 90 min in the vmPFC was also specific to early CS+ offsets, as no significant effect

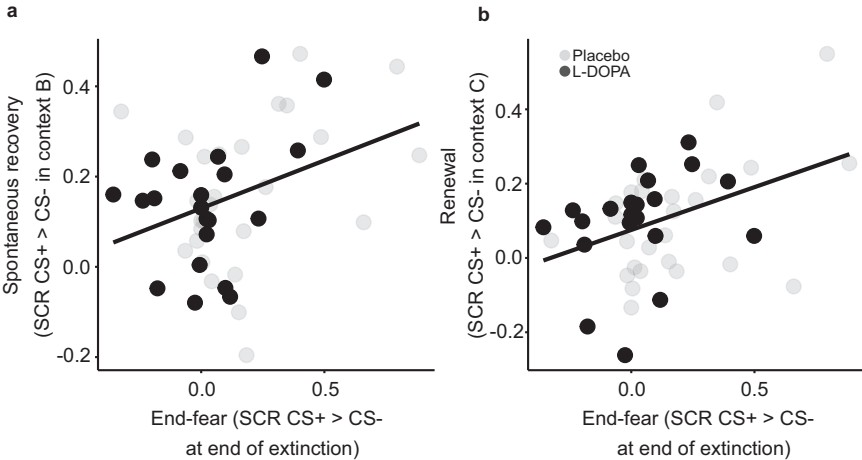

**Fig. 2 | Relationship between CRs at the end of extinction (end-fear) on day 2 and CRs at test in the extinction context B and a new context C on day 3.** Extinction success (assessed by differential CRs (CS + > CS−) at the end of extinction) positively predicted spontaneous recovery (differential CRs (CS + > CS−) at test in extinction context B; (**a**) and renewal (differential CRs (CS + > CS−) at test in a new context; (**b**) in both the placebo- and the L-DOPA-treated groups.

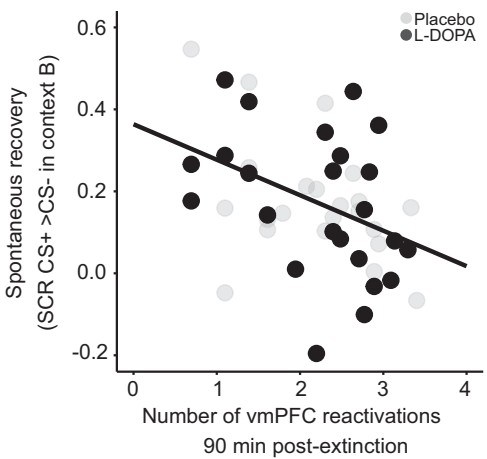

**Fig. 3 | Number of spontaneous reactivations of early CS+ offset -related vmPFC patterns 90 min post extinction on day 2 predicts CRs at test on day 3 in both groups.** Extinction memory retrieval prediction by number of vmPFC reactivations was observed across both L-DOPA- and placebo-treated participants 90 min after extinction. Regression line applies to both groups.

could be detected for early CS- offsets at 90 min in this region (non-preregistered analysis, $\beta = 0.02$, SE = 0.03, $t_{41} = 0.60$, $p = 0.551$).

Together, these results establish an important role for spontaneous post-extinction reactivations of an extinction-related activity pattern in the vmPFC in the consolidation of long-term extinction memories.

The number of vmPFC reactivations at any of the post-extinction time points did not detectably predict differential BOLD responses (contrast CS + > CS−) on day 3. Accordingly, the preregistered analysis of the effects of vmPFC reactivations on task-modulated functional connectivity (contrast CS + > CS−) on this day could not be conducted.

### Effect of L-DOPA administration post extinction on vmPFC reactivations

Next, we tested whether L-DOPA compared to placebo increases the number of spontaneous post-extinction vmPFC reactivations 10, 45, and 90 min after extinction (preregistered hypothesis 3). In congruence with a statistically non-significant effect of L-DOPA on extinction retrieval, there was no statistically significant effect of L-DOPA on the number of vmPFC reactivations during either of the resting-state scans (repeated-measures ANOVA: group: $F_{1,44} = 0.87$, $p = 0.356$, time*group: $F_{1,44} = 0.48$, $p = 0.700$, $n = 46$; Fig. 4). Unplanned Bayesian analysis found strong evidence for H0 ($BF_{01} = 15.00$).

Together, these findings do not confirm an effect of L-DOPA on post-extinction vmPFC reactivations of an extinction-related activity pattern.

### Relationship of salivary alpha-amylase and cortisol with extinction memory retrieval, and interactions with L-DOPA administration post extinction

Our preregistered secondary research question 2 concerned whether individual differences in sAA and sCORT responses to the experimental manipulation on the extinction day 2, involving scanning and an extinction learning session, had an influence on extinction memory retrieval and might moderate the L-DOPA effect. To this aim we collected sAA and sCORT before participants entered the scanner and received extinction training ('baseline', time 0 min in Fig. 1c) and at several time points thereafter on day 2. Time courses of both markers on day 2 exhibited linear decreases (effects of time: sAA: $F_{6,324} = 9.50$, $p < 0.001$, $n = 56$; sCORT: $F_{6,342} = 17.57$, $p < 0.001$, $n = 59$) in the absence of significant group influences (group: sAA:

$F_{1,54} = 0.30$, $p = 0.589$; sCORT: $F_{1,57} = 0.03$, $p = 0.855$; group*time: sAA: $F_{6,324} = 1.71$, $p = 0.119$; sCORT: $F_{6,342} = 0.19$, $p = 0.979$), but did not show the expected peaks at a post-extinction time point relative to baseline (Fig. 5). Maximum levels of sAA and sCORT were observed at baseline in 71.7% and 60.3% of participants, respectively, and at the first post-extinction time point (approx. 25 min in Fig. 1c) in the remaining participants. This suggests the manipulation did not robustly induce arousal and/or stress.

One reviewer pointed out that trait-like sAA activity levels, as apparent from the baseline measurements of sAA on the three experimental days, may also potentially explain individual differences in extinction retrieval and in L-DOPA effects (non-preregistered secondary research question 3). Supporting the notion of trait-like stability in sAA, we observed highly significant medium- and large-sized correlations in baseline sAA between the 3 days (day 1–2: R(51) = 0.46, $p < 0.001$; day 1–3: R(51) = 0.61, $p < 0.001$; day 2–3: R(51) = 0.60, $p < 0.001$; $n = 53$; Supplementary Fig. 4a–c). Test-retest reliability, assessed in a two-way mixed-effects model with participants as random and rater/time as fixed factors, was good (intraclass correlation [ICC] = 0.701, $F_{52,156} = 3.34$, $p < 0.0001$, CI 95% [0.544 0.814]). This allowed us to use average baseline sAA values from the 3 days as signifying an individual's 'trait sAA'. To not neglect potential influences of a current state of the SNS at the time of extinction, as hypothesized in the preregistration, we also defined a 'state sAA' level as the deviation of an individual's maximum value on day 2 from the trait value (thus avoiding collinearity), and entered both into a prediction model for extinction memory retrieval along with factors group and the trait by group and state by group interactions. This calculation of state sAA differed from the preregistration, where we planned to use the difference between the peak and baseline values from day 2 and had not foreseen to calculate a deviation from a trait sAA value. The reason for this change in the analysis procedure was that it was not possible to calculate a peak-baseline difference for the majority of participants, who showed their peak at baseline, and that we had not planned to include trait sAA into the analysis. Note that trait and state values did not significantly differ between groups (two-sample $t$-tests: trait sAA: $t_{57} = 0.21$, $p = 0.836$, CI 95% [−57.94 71.39]; state sAA: $t_{57} = 0.88$, $p = 0.381$, CI 95% [−73.29 28.42]; $n = 59$).

Both trait and state sAA positively predicted CRs at spontaneous recovery (trait sAA: $\beta_{sAA} = 0.0008$, SE = 0.0002, $t_{37} = 2.60$, $p = 0.013$; state sAA: $\beta_{sAA} = 0.0008$, SE = 0.0003, $t_{37} = 2.80$, $p = 0.008$; $n = 43$). The analysis also showed a positive effect of L-DOPA ($\beta_{group} = 0.17$, SE = 0.08, $t_{37} = 2.04$, $p = 0.048$) and a trait sAA by group interaction ($\beta_{sAA*group} = −0.0009$, SE = 0.0004, $t_{37} = 2.13$, $p = 0.040$). There was a non-significant trend-wise state sAA by group interaction ($\beta_{sAA*group} = −0.0008$, SE = 0.0004, $t_{37} = 1.75$, $p = 0.089$). See Fig. 6. That is, higher trait-like or state levels of SNS activity appeared to be associated with relatively impaired extinction learning or consolidation (rather than with improved consolidation, as predicted for peak sAA responses on day 2 by us[28]). More intriguingly, controlling for trait and state sAA revealed a beneficial main effect of L-DOPA on extinction retrieval, and this L-DOPA effect was particularly pronounced at high levels of trait sAA. Whereas trait sAA positively predicted CRs at spontaneous recovery in the placebo group (Fig. 6a), a simple slope analysis detected no such significant sAA-to-spontaneous recovery relationship in the L-DOPA group ($\beta_{placebo} = 0.0006$, SE = 0.0002, $t_{38} = 2.65$, $p = 0.012$; $\beta_{L-DOPA} = −0.0003$, SE = −0.0004, $t_{38} = 0.90$, $p = 0.375$). Hence, the non-preregistered analysis of sAA effects in our study suggests that trait SNS activity is an important moderator of dopaminergic extinction effects, as previously proposed[18,19]. High basal SNS activity may thus be a necessary condition for extinction consolidation augmentation by L-DOPA. The potential moderating influence of state sAA remains open.

A further non-preregistered exploratory analysis revealed that trait (but not state) sAA was significantly associated with high

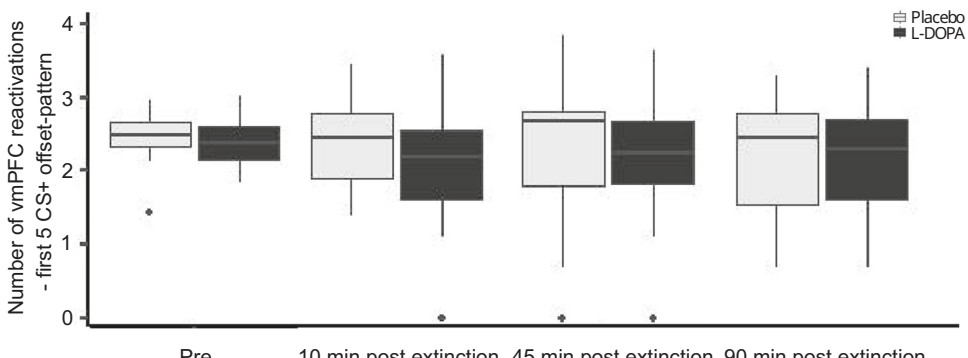

**Fig. 4 | Effect of post-extinction L-DOPA administration on the number of spontaneous reactivations of CS+ offset-related vmPFC patterns during resting-state scans on day 2.** There was no effect of L-DOPA on numbers of vmPFC reactivation patterns during resting-state fRMI 10, 45, and 90 min post extinction learning (n = 46). Box plots indicate 1st, 2nd, and 3rd quartile with a median center line; whiskers indicate ±1.5*IQR. Dots represent outliers lying outside this range.

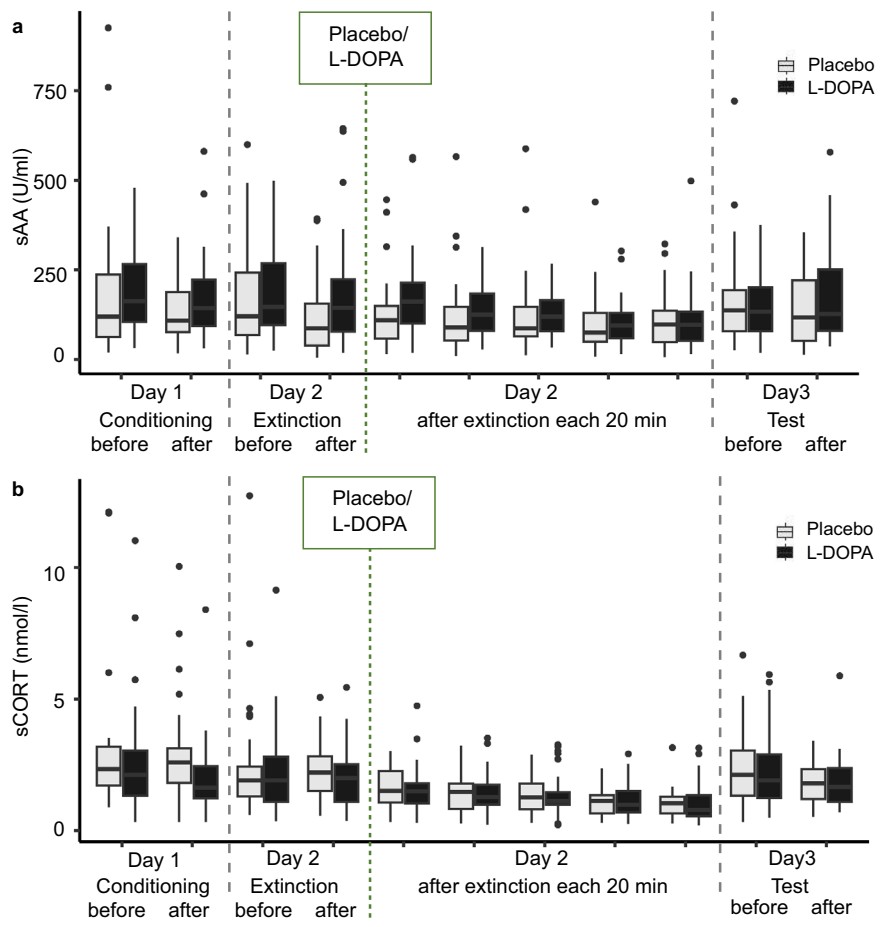

**Fig. 5 | sAA and sCORT measurements on days 1–3. a** sAA and **b** sCORT levels exhibited linear decreases on day 2 with peak levels in the group observed before extinction. sAA and sCORT did not differ on day 1 between time points and groups, that is, before and after conditioning (sAA: time: $F_{1,59} = 2.28$, $p = 0.136$, group: $F_{1,59} = 0.72$, $p = 0.401$, time*group: $F_{1,59} = 0.88$, $p = 0.352$, $n = 61$; sCORT: time: $F_{1,64} = 2.95$, $p = 0.091$, group: $F_{1,64} = 2.78$, $p = 0.100$, time*group: $F_{1,64} = 0.28$, $p = 0.601$, $n = 66$). On day 3, only sCORT was significantly decreased after the experimental procedure in both groups (sAA: time: $F_{1,54} = 0.26$, $p = 0.616$, group: $F_{1,54} = 0.29$, $p = 0.593$, time*group: $F_{1,54} = 2.97$, $p = 0.091$, $n = 57$; sCORT: time: $F_{1,57} = 10.67$, $p = 0.002$, group: $F_{1,57} = 0.07$, $p = 0.798$, time*group: $F_{1,57} = 0.18$, $p = 0.673$, $n = 60$). Saliva samples were collected as shown in Fig. 1c. Box plots indicate 1st, 2nd, and 3rd quartile with a median center line; whiskers indicate ±1.5*IQR. Dots represent outliers lying outside this range.

extinction end-fear, that is, poor extinction success (trait sAA: $\beta_{sAA} = 0.002$, SE = 0.0008, $t_{33} = 2.53$, $p = 0.016$; state sAA: $\beta_{sAA} = -0.001$, SE = 0.0007, $t_{33} = 1.63$, $p = 0.112$; group: $\beta_{group} = -0.04$, SE = 0.15, $t_{33} = 0.24$, $p = 1.000$; $n = 39$; Supplementary Fig. 5). These results may support generally impairing effects of trait-like SNS activity on extinction, as suggested by animal studies[39], which in turn might be carried over to spontaneous recovery (see results above). There were no statistically significant effects of trait or state sAA on extinction retrieval in the renewal test (non-preregistered exploratory analysis, not shown).

For baseline sCORT, we did not observe temporal stability comparable to baseline sAA (day 1–2: R(55) = 0.53, $p < 0.001$, day 1–3: R(55) = 0.32, $p = 0.015$, day 2–3: R(55) = 0.29, $p = 0.027$; ICC = 0.03, $F_{56,168} = 1.03$, $p = 0.429$, CI 95% [−0.458 0.385]; $n = 57$; Supplementary Fig. 4d–f). We nevertheless for completeness also tested the effects of trait and state sCORT on extinction retrieval and L-DOPA action as well on extinction success. Trait and state sCORT did not differ significantly between groups (trait sCORT: $t_{61} = 0.87$, $p = 0.386$, CI 95% [−0.46 1.17]; state sCORT: $t_{61} = 0.06$, $p = 0.955$, CI 95% [−0.60 0.57]; $n = 63$). There were no significant effects.

### Relationship of salivary alpha-amylase with vmPFC reactivations during memory consolidation, and interaction with L-DOPA administration post extinction

As controlling for trait and state sAA in the present study had revealed the hypothesized beneficial effect of L-DOPA on extinction memory retrieval (which had not been found when testing for an L-DOPA effect alone), we generated the new hypothesis that controlling for sAA might also reveal a beneficial effect of L-DOPA on vmPFC reactivations 90 min after extinction. This non-preregistered exploratory analysis found significant state sAA ($\beta_{sAA} = -0.003$, SE = 0.002, $t_{43} = 2.04$,

$p = 0.047$, $n = 49$) and state sAA by group effects ($\beta_{sAA*group} = 0.005$, SE = 0.003, $t_{43} = 2.04$, $p = 0.048$). See Fig. 7. There were non-significant trend-wise effects of trait sAA ($\beta_{sAA} = -0.002$, SE = 0.001, $t_{43} = 1.86$, $p = 0.069$), group ($\beta_{group} = 0.83$, SE = 0.002, $t_{43} = 1.88$, $p = 0.067$), and trait sAA by group ($\beta_{sAA*group} = 0.004$, SE = 0.002, $t_{43} = 1.79$, $p = 0.081$). This suggests that, unlike in the case of extinction memory retrieval, the effect of state sAA may be important when investigating consolidation-related vmPFC activity.

To qualify the state sAA by group interaction, a simple slope analysis showed a non-significant tendency for a negative influence of state sAA on vmPFC reactivations in the placebo group ($\beta_{placebo} = -0.003$, SE = 0.002, $t_{43} = 2.02$, $p = 0.050$). The state sAA-to-vmPFC reactivations relationship tended to be inversed in the L-DOPA group ($\beta_{L-DOPA} = 0.002$, SE = 0.002, $t_{43} = 0.96$, $p = 0.340$; Fig. 7b). That is, higher apparent current SNS activity was associated with fewer reactivations in the placebo than in the L-DOPA group. This latter result resembles in its pattern the result shown in Fig. 6a of an interaction between trait sAA and L-DOPA in their influence on spontaneous recovery.

There were no significant effects when analyzing reactivations at 45 min after extinction.

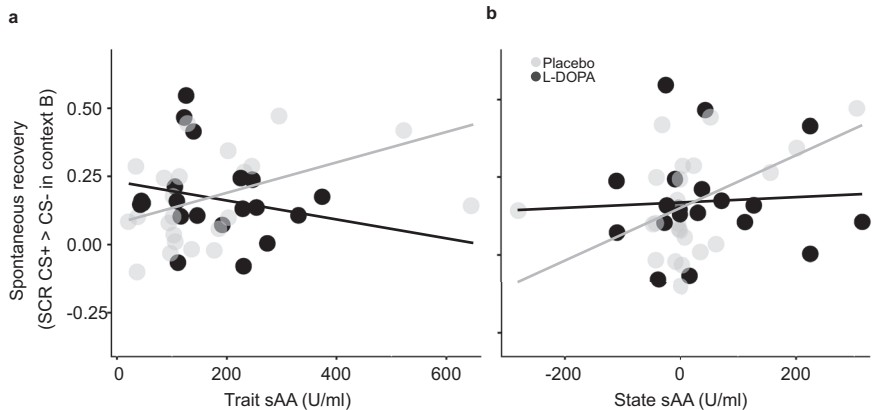

**Fig. 6 | Relationship between trait and state sAA and CRs at test in context B on day 3. a** Whereas high trait sAA was predictive for impaired extinction memory retrieval at test in the extinction context B (spontaneous recovery) in the placebo group, this effect was rescued by L-DOPA administration. **b** A similar, though non-significant pattern emerged also in participants with high state sAA on day 2, defined as deviation of the individual's peak sAA level on that day from their trait sAA value. Regression lines shown represent the predicted relationship between trait or state sAA and CRs at test based on the implemented model.

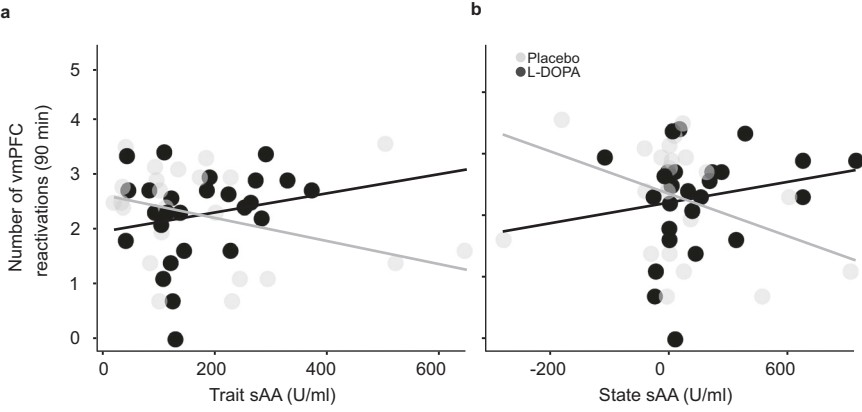

**Fig. 7 | Relationship between trait and state sAA and number of vmPFC reactivations 90 min after extinction learning. a** High trait sAA was non-significantly associated with vmPFC reactivations 90 min after extinction. **b** High state sAA showed a negative association with vmPFC reactivations 90 min after extinction in the placebo group, whereas this effect was not statistically significant in the L-DOPA group. Regression lines shown represent the predicted relationship between trait or state sAA and CRs at test based on the implemented model.

## Discussion

The present study aimed to replicate the main study results of Gerlicher et al.[16]. We tested whether post-extinction L-DOPA as compared to placebo administration would improve extinction memory retrieval in healthy normal male participants, as visible from reduced spontaneous recovery during an extinction memory test on day 3 of our experimental paradigm (hypothesis 1); whether the number of spontaneous reactivations of an extinction learning-related multi-voxel pattern (MVP) observed in the vmPFC in post-extinction resting-state scans would be predictive for extinction memory retrieval across treatment groups (hypothesis 2); and whether L-DOPA administration would relatively increase post-extinction vmPFC reactivations (hypothesis 3). The latter was done with an eye on potentially finding the mediation of the L-DOPA effect on retrieval by the increased number of vmPFC reactivations that was reported by Gerlicher et al.[16]. Two preregistered extensions further aimed at clarifying whether L-DOPA administration protects against the renewal of fear, tested on day 3 by presenting the CSs in a non-extinction context C (secondary research question 1), and whether peak sAA or sCORT responses to the experimental manipulation (scanning, extinction) on the extinction day 2 predict extinction memory retrieval on day 3 and interact with drug treatment in doing so (secondary research question 2). Upon request of a reviewer, we also tested whether trait-like baseline activity levels of sAA (and for completeness, trait-like baseline concentrations of sCORT) showed predictive effects (non-preregistered secondary research question 3).

We confirmed one of our three main hypotheses, namely that the number of spontaneous post-extinction reactivations of a vmPFC activity pattern observed during CS+ offsets in early extinction is predictive for extinction memory retrieval (hypothesis 2). Specifically, this was observed for the resting-state scan conducted 90 min after extinction, though not at 45 min as in the discovery study. We could not replicate the facilitating effect of L-DOPA on extinction retrieval (hypothesis 1) and on the number of vmPFC reactivations (hypothesis 3). Examining whether L-DOPA could make extinction memories context independent, we also did not find an advantage of L-DOPA administration for extinction memory retrieval in a renewal test (secondary research question 1). A sensitivity analysis revealed that our study had the power to detect effect sizes equal to or larger than $\eta_p^2 = 0.035$ (f = 0.192). This means that our study was well-equipped to detect moderate to large effects but may have limitations in detecting very small effects that account for less than 3.5% of the variance. As effect sizes might be inflated, an effect size half of our previously discovered effect size[16] ($\eta_p^2 = 0.166$) still could have been detected. Answering secondary research question 2 was complicated by absence of a clear response in sAA and sCORT measures to the experimental manipulation. Contrary to our expectations, being placed in a scanner, experiencing scanner noise, and being subjected to an extinction training session did not lead to detectable increases at the group level in these state markers of SNS and HPA activity, which would have been indicative of an acute arousal and stress response. We, therefore, focused our analysis on the non-preregistered secondary research question 3, centered around potential predictive effects of trait-like sAA and sCORT levels. Trait sAA and sCORT were operationalized as average baseline values from all three experimental days, while potential influences of current activity levels were assessed by entering the deviation of individuals' maximum sAA and sCORT levels on day 2 from their respective trait levels into a common prediction model with treatment group and the trait by group and state by group interaction terms. These analyses found that trait and state sAA were negatively associated with extinction memory retrieval. Further, trait sAA (but not state sAA) levels interacted with drug treatment in that the generally negative association of trait sAA with extinction retrieval (higher spontaneous recovery) did not reach statistical significance in the L-DOPA group. Unexpectedly, the analysis also showed a beneficial main effect of L-DOPA on extinction retrieval. Hence, controlling for

trait and state sAA levels revealed an L-DOPA effect that was not significant in the uncontrolled analysis conducted to test hypothesis 1.

We complemented these main analyses with an exploratory analysis of trait and state sAA on extinction learning success on day 2, finding that extinction learning is poor in individuals with high trait sAA, and with an exploratory analysis of L-DOPA effects on vmPFC reactivations 90 min after extinction that took into consideration trait and state sAA. Here, state sAA negatively predicted reactivations and state sAA and L-DOPA treatment interacted in the sense that state sAA was negatively related with the number of vmPFC reactivations in the placebo group, but tended to be positively related in the L-DOPA group.

Our main replication finding is that spontaneous post-extinction reactivations of a CS+ offset-related vmPFC activity pattern positively associate with later extinction memory retrieval. This result is in line with previous animal research, showing that spontaneous IL activity after extinction learning is crucial for consolidation and predictive for memory retrieval[49]. Additionally, stimulus-specific fMRI multi-voxel reactivation patterns in other learning domains and brain regions also have been reported to be associated with later memory performance outcomes[50]. The reactivation pattern in our work specifically recapitulates CS+ offset-related activity in the vmPFC during early extinction. The CS+ offset is the time point in a trial at which participants have learned to expect a US during prior conditioning, and the omission of the US that occurs during the extinction phase is most unexpected at the beginning of extinction, while less of a surprise at later extinction trials. This suggests that the vmPFC may reactivate a prediction error (PE) signal, which learning theory considers to be the key teaching signal in associative learning[22,51]. In extinction specifically, the PE may not so much lead to an update (reduction) of the aversive value of the CS, but rather to the build-up of a new, appetitive safety memory (a CS−noUS association, or extinction memory) that antagonistically inhibits the CS-US association (or fear memory)[52,53]. On this basis, one can postulate that vmPFC reactivations of important extinction learning events after extinction provide the link between extinction learning and extinction memory retrieval, by facilitating the storage and consolidation of a safety association.

One open question is why we find a predictive effect of vmPFC reactivations for memory retrieval at 90 min after extinction and not at 45 min, as in the discovery study. Animal studies indicate that the time window for consolidation processes can be broad, spanning hours and even extending into sleep, depending on which molecular or network mechanisms are involved[54]. Further, while molecular consolidation processes are bound to specific timelines[55], they may still show substantial interindividual differences, for instance, due to differences in the speed or success of learning, in the individual make-up of the molecular systems, or in the current state of the system in which a consolidation process occurs (e.g., in stress, vigilance, arousal etc.). Such differences have not been explored in human work but may lead to substantial heterogeneity in consolidation dynamics across studies. It is also worth noting that oral administration of a drug can lead to substantial pharmacokinetic variability.

Previous studies from our group using a cue conditioning and extinction paradigm have reported pronounced L-DOPA main effects when experiments were conducted in MRI[14,16], but only conditional L-DOPA effects (in participants with successful extinction learning) when experiments were purely behavioral (two experiments[18]). Because there is evidence that the scanner environment is frequently perceived by participants as arousing and stressful[36–38], this pattern of results prompted us to postulate that L-DOPA might be more effective in promoting extinction memory consolidation under conditions of high arousal and/or stress, as should be apparent from high sAA or sCORT responses on the extinction day[28].

Our analysis of sAA and sCORT time courses found no evidence of such responses at the group level. The same was observed on the other

experimental days (conditioning, retrieval test; see Fig. 6). The absence of a clear cortisol response is not very surprising, given that cortisol is typically not increased by conditioning experiments in humans[56]. The absence of sAA responses could potentially be attributed to a habituation effect, whereby the initial peak of sAA activity might not have been captured, since the time participants spent in the scanner each day before the second saliva sampling was at least 20 min. Another part of the explanation may lie in the limited reliability of sAA as a reactivity marker[57].

Unexpectedly, however, our exploratory analyses revealed a pronounced predictive effect of trait sAA on extinction memory retrieval and on the influence of L-DOPA treatment on this outcome. sAA activity levels repeatedly measured in states of rest show high temporal stability, higher between- than within-subject differences, and substantial heritability[44,45,47,48]. To establish trait sAA levels, it is recommended to assess sAA on two or three days, whereby the number of days is more important than the number of measurements per day[44]. This here allowed us to use average baseline sAA values from three experimental days to obtain a trait-like individual-differences marker.

Baseline, or resting, sAA is lowered by systemic administration of the beta-adrenergic receptor blocker propranolol[58,59] and raised by the alpha2-adrenergic antagonist yohimbine, which also increases plasma noradrenaline levels[60], indicating a direct link between baseline sAA and noradrenergic activity and, by extension, the basal activity of the sympathetic nervous system (SNS). The suitability of baseline sAA as a marker for basal SNS activity is further supported by a systematic positive link between baseline sAA and age[61-63], which is in line with the established finding that older adults have higher chronic SNS activation[64], and by the observation that morning sAA levels are associated with hypertension[63].

The same study showed a positive relationship between morning sAA and Mild Cognitive Impairment, while another study showed a relationship between morning sAA and cognitive impairments in younger adults[65]. Baseline sAA has also been linked with poor pattern separation[66]. These data indicate a negative relationship between baseline sAA and cognition and suggest that this measure may be usable as a marker for neural function beyond basal SNS activity. The apparent link between baseline sAA and cognition may be explained via their common relationship with general noradrenergic activity, based on the finding that higher noradrenaline levels in cerebrospinal fluid are associated with worse cognition[67].

Taken together, the literature on baseline sAA indicates that trait-like sAA values from baseline states reflect basal SNS activity and index poorer cognitive performance. This in turn makes it a plausible extension of the current knowledge that the trait-like individual differences in baseline sAA activity observed in our study are a predictor of individual differences in extinction memory performance.

Trait sAA in our data predicted poorer extinction memory retrieval, poorer extinction success, and (non-significantly) fewer post-extinction vmPFC reactivations.

The negative association with extinction retrieval did not reach statistical significance when L-DOPA was administered after extinction. This finding suggests that L-DOPA is more effective in participants with high basal SNS activity. From a clinical point of view, this points to an important boundary condition for a potential application of L-DOPA as a pharmacological augmentation of exposure-based therapy[9]. L-DOPA-based augmentation may be specifically indicated in patients in which a prior determination of trait sAA levels has revealed high basal arousal. From a scientific point of view, we may have identified an important individual-differences factor that should be considered in future studies testing methods to optimize extinction or exposure therapy – via L-DOPA and potentially also via other routes.

The present data are not conclusive as to whether current sAA levels (state sAA) may be of similar importance as trait sAA. Although we did observe a negative predictive effect of state sAA for extinction retrieval as well as vmPFC reactivations, there was only a non-significant trend-level interaction with L-DOPA treatment on retrieval. Because state sAA in our analysis was defined in relation to individuals' trait sAA levels, an independent influence of current sAA activity remains elusive. Future studies will in any case have to test the combined influence of both variables.

Mechanistically, a simple explanation for an apparently better efficacy of L-DOPA with high basal SNS activity may be that L-DOPA has more room to improve extinction consolidation when it is compromised by (nor)adrenergic activity. Alternatively, L-DOPA may also directly interfere with the impairing effects of central noradrenaline on extinction consolidation. Previous research has suggested that, at high arousal levels, noradrenaline released from locus coeruleus (LC) projections in the basolateral amygdala (BLA) acts via α1- and β-adrenoceptors to increase BLA activity, while noradrenaline release in the mPFC reduces activity there. Further, the increased BLA activity also attenuates mPFC output via inhibitory BLA-mPFC projections. In sum, there is an increase in fear and impeded learning and/or consolidation[33,39]. Dopamine may counteract these impairing effects of noradrenergic activity, first, by decreasing prefrontal extracellular noradrenaline levels via increased neuronal reuptake and activation of inhibitory α2-adrenoceptors[13], and second, by directly reducing BLA-mediated inhibition of the mPFC[68]. Together, these findings point to a possible mechanism by which systemic L-DOPA administration reduces the detrimental effects of high arousal on extinction learning and consolidation.

It is important to point out that our findings of trait and, to a lesser extent, state sAA influences on extinction and L-DOPA-based extinction augmentation result from non-preregistered analyses and that further investigation is needed to replicate them. Importantly, future work should also experimentally manipulate the SNS. We mention here that self-reported trait and state anxiety did not statistically significantly correlate with trait or state sAA in our data (Supplementary Table 5). The concept of arousal is multifaceted, encompassing subjective, behavioral, and physiological components, and response systems can be desynchronized[69,70]. Our results suggest that trait sAA is a relevant factor for the facilitation of consolidation with L-DOPA. A further limitation is that our method to collect salivary samples for sAA analysis could be improved, in particular by utilizing collection devices with synthetic swabs or other appropriate materials instead of cotton swabs. Beyond the role of salivary measures, it should be noted that all reported effects on extinction retrieval in this study were observed in skin conductance responses as our main index of conditioned responding. There were no predictive relationships of vmPFC reactivations, sAA, or drug treatment on US expectancy ratings. SCRs are implicit and objectively measurable, but expectancy or other ratings have the practical advantage that they can be easily collected, in particular also in clinical settings. One explanation for the absence of statistically significant effects on ratings may lie in fear responses being carried by dissociable systems (e.g., ref. 71), not all of which may be affected by our manipulation. Finally, a clear gap in the current extinction and L-DOPA literature is that findings have not yet been extended to women.

In this replication study, we aimed at reproducing the main findings from Gerlicher et al.[16] with minor extensions. Our results establish the predictive role of a potentially PE-related reactivation pattern in the vmPFC for extinction memory retrieval, tested in spontaneous recovery. In our exploratory analyses, we discover that high basal SNS activity in male humans is associated with impaired extinction consolidation and that L-DOPA promotes extinction consolidation specifically in individuals with high basal SNS activity. This result opens up new avenues for the investigation of L-DOPA as an enhancer of exposure therapy, while also emphasizing the need for future confirmatory studies. For fear extinction research, our results lead to new mechanistic hypotheses and

suggest a new possibility to explain the vast individual differences frequently observed in human extinction research.

## Methods

The experiment was approved by the local ethics committee (Ethikkommission der Landesärztekammer, Rhineland-Palatinate, Germany) and was conducted in accordance with the Declaration of Helsinki.

### Preregistration

The protocol of this registration study can be found under the following link: https://osf.io/x64cn/ (submitted 10/19/2018).

### Design

The study design is as in the discovery study, with the exception of the added sAA and sCORT measurements on days 1 and 2 and the added renewal test on day 3.

### Technical parameters

All technical parameters, including the MRI scanner and equipment, were kept identical to those used in the initial study.

### Sample size

Our major effect of interest in the discovery study[16] was the interaction effect of stimuli (CS + /CS−) and group (placebo/L-DOPA) on average SCRs during the test in the extinction context on day 3, corresponding to our replication hypothesis 1. This effect had a size of $\eta_p^2 = 0.166$, calculated by IBM SPSS Statistics (version 23, Chicago IL). Using this effect size, we estimated using G*power 3.1.9.2[72] that with a power of 80% and an alpha of 0.05, the required sample size would be $N = 22$, i.e., $n = 11$ participants per group. As stated in the Introduction, further statistical, logistical, and ethical considerations led us to aim at a sample size of $N = 63$ after drop-outs, requiring us to recruit 70 participants. Supplementary Table 1 gives demographic information and further characteristics of the full recruited sample.

Technical problems in data acquisition, described below for each data modality in the corresponding Methods sections, reduced sample sizes for the analysis of the data modalities. Supplementary Table 2 lists the missing data per participant, data modality, and day. The final n for each analysis is indicated in the corresponding sections of the Results. For the testing of our main hypothesis 1, using SCR data, we achieved a final n of 55, which is 1.4 times greater than the discovery sample and 2.5 times greater than the critical sample size estimated based on power calculation. We further implemented a sensitivity analysis using G*power 3.1.9.4 to test whether the study was sufficiently powered to detect the smallest theoretically or pragmatically meaningful effect.

### Participants

As in the discovery study, we restricted recruitment to individuals of male sex. The estrous cycle has been shown to interact with extinction memory consolidation[73,74], and dopamine has been shown to have opposing effects on extinction depending on estrous cycle phase[75]. Thus, by limiting ourselves to male participants, we could expect higher sample homogeneity and correspondingly higher chances for detecting a neurobiological mechanism. After safely demonstrating an effect with this strategy, a necessary next step is to test transfer to non-male populations. A board-certified physician screened participants for contraindications of L-DOPA intake, current physiological, neurological, or psychiatric disorders, excessive consumption of nicotine (>10 cigarettes/day), alcohol (>15 glasses of beer/wine per week), or cannabis (>1 joint/month), participation in other pharmacological studies, and skin conductance non-responding (assessed by eSense Skin response, Mindfield® Biosystems Ltd., Berlin, Germany). Drug abuse was assessed via a urine test (M-10/3DT; Diagnostik Nord, Schwerin, Germany). All participants gave informed consent. After experiment completion, participants were reimbursed with 120 Euros.

### Experimental design

We used a 3-day fear conditioning and extinction paradigm consisting of conditioning (day 1) in context A, extinction (day 2) in context B, and tests for the effect of L-DOPA on extinction memory retrieval in the original extinction context (B) and in a new context (C) (day 3). See Fig. 1a. Our study involves a $2 \times 2$ mixed factorial design with stimulus as within- (CS+ vs. CS−) and drug group as between-subject (placebo vs. L-DOPA) factor. As in our previous studies[14,16,18,19], the main outcome measures were average CS evoked SCRs during each of the test contexts on day 3.

### Stimuli

Two black geometric symbols (a square and a rhombus) presented in the center of the screen served as CSs. The symbols were superimposed on background pictures of one of three different gray photos (living room, kitchen, and sleeping room), which served as contexts A, B, and C. The assignment of symbols to the CS+ and CS- and backgrounds to the conditioning, extinction or renewal context were randomized between participants and groups. Diminishing the risk of low-visual feature differences between the CS+ and the CS−, contrast and luminance of stimuli were adjusted using SHINE toolbox[76]. Stimuli were presented using Presentation Software (Presentation®, Neurobehavioral Systems, Inc., Berkeley, CA, USA). A painful electrical stimulation consisting of three square-wave pulses of 2 ms (50 ms interstimulus interval) was employed as US. Pain stimuli were generated by using a DS7A electrical stimulator (Digitimer, Weybridge) and delivered to the skin through a surface electrode with a platinum pin (Specialty Developments, Bexley, UK). Due to observed incidences of high-voltage MRI artefacts in previous studies, we moved the stimulation further away from the magnet's bore from the dorsal hand (preregistered location of stimulation) to the ankle.

### Drug treatment

Participants were randomly assigned to the L-DOPA or the placebo group using a randomization list generated before the start of the study, with the restriction that groups had to be matched on self-reported trait anxiety based on the State-Trait Anxiety Inventory questionnaire (STAI-T[77]). STAI-T scores did not differ between groups after acquisition of $n = 45$ participants, and therefore the predefined treatment group randomization order was kept. After full data acquisition, STAI-T values did not differ between groups (Supplementary Table 1). Other than in ref. 16 and than preregistered[28], anxiety sensitivity index (ASI) scores, originally intended to also be matched between groups, were not acquired due to an initially undetected technical failure. Drug preparation was done by a person not involved in the experiments or analyses. Participants were administered either 150/37.5 mg L- DOPA-benserazide (Levodopa-Benserazid-ratiopharm®, Germany; for dosage see refs. 14,17) or a visually identical capsule filled with mannitol and aerosil (i.e., placebo). Drugs were prepared and provided by the pharmacy of the University Medical Center Mainz and administered in a double-blind fashion. Participants were asked to refrain from eating, consuming caffeinated drinks, and smoking 2 h prior to drug intake. Fasting L-DOPA absorption is rapid, with L-DOPA peak plasma concentration reaching within 15 to 60 min after oral intake[78]. Furthermore, L-DOPA half-time is also short (90 min), excluding that drug effects on extinction retrieval can be explained by direct drug action on the test day.

### Experimental procedures

**Day 1 - fear learning.** Participants filled out the state version of the state-trait anxiety inventory (STAI-S), the Acceptance and Action Questionnaire (AAQ[79]), the General Health Questionnaire (GHQ[80]) (no group differences in all questionnaires; for analyses, see Supplementary Table 1) and answered a list of questions assessing behaviors potentially influencing on sCORT measurements (no group differences

in all measures; for analyses, see Supplementary Table 3). Subsequently, a saliva sample was collected. Participants were placed in the MRI scanner, and SCR and pain stimulation electrodes were attached. An 8-min resting-state scan was conducted. Participants then were familiarized with the experiment through two CS presentations in each of the three contexts and a training of US-expectancy ratings (for analyses, see Supplementary Fig 1). US-expectancy ratings were taken on all three days as additional check for the success of conditioning, extinction, and extinction retrieval, respectively. Subsequently, US-intensity was calibrated to a level rated as "maximally painful, but still tolerable" (for analyses, see Supplementary Table 1). Participants were then instructed that one symbol would never be followed by a pain stimulus and that their task was to find out what rule applied to the other symbol. After scanning onset, the paradigm started with US-expectancy ratings for each CS. After the initial rating, a background picture representing context A appeared on the screen. The context picture remained on the screen continuously throughout conditioning. Participants were presented with 10 CS+ and 10 CS− trials. Notably, 5 out of 10 CS+ presentations (i.e., 50%) were reinforced. CSs were presented for 4.5 s. In case of reinforced CS+ presentations, the US was delivered such that it co-terminated with the CS presentation. Inter-trial intervals (ITIs) lasted 17, 18, or 19 s. Trial order was randomized in such a way that not more than two trials of the same type (i.e., CS+ with US, CS+ without US, CS−) succeed each other. Conditioning lasted approximately 12 min and scanning ended after picture offset and US-expectancy ratings. After conditioning, another 8-min resting-state scan was conducted, followed by anatomical scans (T1, T2, DTI, see Acquisition of MRI data). Subsequently, electrodes were detached, participants gave another saliva sample and filled out a list of questions assessing contingency knowledge (for details, see Supplementary Methods). The whole procedure lasted approximately 90 min.

**Day 2 - Extinction learning and consolidation.** After 24 h (±2 h), participants came back to the laboratory to fill out the STAI-S questionnaire and answer questions on behaviors potentially influencing sCORT measures. Participants provided a saliva sample and were placed in the scanner. Electrodes were attached, and participants were instructed that the experiment would continue, and that their individual US strength from day 1 would be applied. An 8-min resting-state scan was conducted. Before and after extinction, US-expectancy ratings were taken. During extinction, a background picture representing context B was continuously shown on the screen and participants were presented 15 CS+ and CS− trials, using the same timings and pseudo-randomization algorithm as in conditioning. Extinction lasted approximately 15 min. Subsequently, participants were taken out of the scanner for saliva sample collection, detaching of electrodes and receiving either a placebo or L-DOPA pill. Participants stayed under observation for 90 min. During this period, further 8-min resting-state scans were performed -10, 45, and 90 min after extinction end, and saliva samples were collected each 20 min starting from the saliva collection time point at extinction end (see Fig. 1c). Before leaving the laboratory, participants filled out the STAI-S, a list of questions assessing contingency knowledge and a questionnaire assessing possible side-effects of L-DOPA intake (for details, see Supplementary Table 4). The whole procedure lasted approximately 150 min.

**Day 3 – Extinction memory retrieval test (spontaneous recovery and renewal).** After 24 h (±2 h), participants came back to the laboratory and filled out the STAI-S and side-effects questionnaires and answered questions regarding potential confounding factors on sCORT measurements. Participants provided a saliva sample and were placed in the scanner, electrodes were attached, and participants were instructed that their US strength from day 1 would be applied and that the experiment would continue. An 8-min resting-state scan was

conducted. Before and after the test, US-expectancy ratings were taken. During the test, participants were then presented 10 CS+ and 10 CS− trials in context B first (spontaneous recovery test) and subsequently, the same number of trials in context C (renewal test), applying timings and randomization as on day 1. Finally, another saliva sample was taken, and participants filled out a questionnaire assessing contingency knowledge. The test lasted about 20 min; the whole procedure lasted approximately 60 min.

**Skin conductance responses (SCRs).** CS elicited SCRs were employed as conditioned fear responses (CRs). Electrodermal activity was recorded from the thenar and hypo-thenar of the non-dominant hand using self-adhesive Ag/AgACl electrodes prefilled with an isotonic electrolyte medium and the Biopac MP150 with EDA100C device (EL-507, BIOPAC® Systems Inc., Goleta, California, USA). The raw signal was amplified and low-pass filtered with a cut-off frequency of 1 Hz. The onset of SCRs was visually scored offline in a time window from 900 to 4000 ms after CS onset. The phasic amplitude of SCRs was calculated by subtracting the onset background tonic skin conductance level (SCL) from the subsequent peak, using a custom-made analysis script. Technical problems in data acquisition led to missing data in $n = 0/3/3$ participants for days 1/2/3, respectively. SCRs with amplitudes smaller than 0.02 μs were scored as zero and remained in the analysis. If more than 75% of trials during an experimental session were scored as zero, data of this participant during that session was regarded as invalid and excluded from SCR analysis ($n = 4/16/12$ on days 1/2/3). Hence, in total, data from $n = 4/19/15$ participants on days 1/2/3 were not available for analysis. To normalize distributions, data was log-transformed (+1 and log) and range-corrected for each participant and experimental session (i.e., (SCR - SCRmin) / SCRmax[81]).

**Heart rate.** Heart rate was recorded with an MRI-compatible fiber-optic pulse oximeter during both resting-state scans and experimental sessions. We assessed heart rate during the resting-state phases preceding each experimental session as an indicator of autonomic stress response. Further analyses were not implemented, as data was largely lost due to a storage error.

**Pupil dilation.** Pupil dilation was assessed monocularly using an MRI-compatible camera (MR Cam Model 12 M; MRC systems, Germany). The iViewX 2.8 software (Sensormotoric Instruments, Germany) was used for recordings. Contrary to our pre-registration, no analyses were conducted, as data was largely incomplete due to technical camera problems.

**US-expectancy ratings.** Participants were asked to indicate the expectancy of receiving an electric pain stimulation for each CS with a cursor on a visual analogue scale from 0 = "no expectancy" to 100 = "high expectancy". The start position of the cursor on the scale was determined randomly. Ratings were not available for $n = 0/3/3$ participants on days 1/2/3. According to the ratings, participants showed successful fear acquisition, extinction, and extinction memory retrieval. For results, see Supplementary Fig 1.

**Salivary alpha-amylase and cortisol measurement.** To account for potential fluctuations in salivary cortisol levels, all experimental procedures were scheduled to commence no earlier than 4 pm. This time allocation was implemented to ensure a balanced representation of cortisol levels across participants and minimize any confounding effects that may arise from diurnal variations. Saliva samples were collected from the participants outside the scanner using cotton swabs. Samples were taken before and after the fMRI scan on days 1 and 3. On day 2, the saliva samples were collected before and after the first fMRI scan. Five additional saliva samples were collected approximately every 20 min after extinction learning (total $n = 11$, Fig. 1c for

timeline). Saliva sample collection followed a standardized procedure using Salivettes with cotton swab and white cap from the Sarstedt manufacturer. Participants were instructed to place the swab in their cheek without chewing on it. The swab remained in the cheek timed for one minute to allow for sufficient saliva collection. Samples were collected throughout each day's experiment, and they were stored in the fridge at −20 °C approximately 10 min after the experiment's completion. The elapsed time from the first sample collection (baseline) to freezing was approximately 100 min for day 1, 160 min for day 2, and 70 min for day 3. All samples remained frozen throughout the data collection period of 3 years, and they were thawed 1 day prior to sending them to the analyzing laboratory. All samples were analyzed within 2 weeks after being sent to the laboratory. sCORT and sAA were determined using a commercial enzyme immunoassay, see Supplementary Methods. Data was missing due to lack of saliva, or incomplete due to insufficient saliva for more than one measure from one swab, that is, sCORT was measured first, and no more saliva was left for the sAA measurement (missing values: sCORT: $n = 4$ on day 1, $n = 5$ on day 2 pre extinction, $n = 11$ of six swab times post extinction, $n = 10$ on day 3; sAA: $n = 9$ on day 1, $n = 6$ on day 2 pre extinction, $n = 14$ of six swab times post extinction, $n = 14$ on day 3).

**Acquisition of MRI data.** MRI data was acquired on a Siemens MAGNETOM Trio 3 Tesla MRI System using a 32-channel head coil. Resting-state and task data were recorded using gradient echo, echo planar imaging (EPI) with a multiband sequence covering the whole brain (TR: 1000 ms, TE: 29 ms, multi-band acceleration factor: 4, voxel-size: 2.5 mm isotropic, flip angle 56°, field of view: 210 mm[82]). A high-resolution T1 weighted image was acquired after the experiment on day 1 for anatomical visualization and normalization of the EPI data (TR: 1900 ms, TE: 2540 ms, voxel size: 0.8 mm isotropic, flip angle 9°, field of view: 260 mm, MPRAGE sequence). T2 weighted images were collected for preventative neuro-radiological diagnostics for all participants (45 slices, TR: 6100 ms, TE: 79 ms, voxel size: 3 mm isotropic, flip angle: 120°, Turbo Spin Echo (TSE) sequence). Lastly, we collected multidimensional diffusion-weighted tensor images (DTI) from each participant (72 slices, voxel-size: 2 mm isotropic, TR: 9100 ms, TE: 85 ms, number of directions: 64, diffusion weights: 2, b-value 1: 0 s/mm², b-value 2: 1000 s/mm², Multi-Directional Diffusion Weighted (MDDW) sequence).

For the acquisition of the resting-state scans, participants were instructed to remain awake, keep their eyes open, fixate a black cross presented in the center of a gray screen, and let their mind wander freely. Compliance with the instruction to remain awake was monitored online using video recordings of the right eye. After each resting-state scan, participants rated their tiredness on a scale from 0 = "not tired at all" to 100 = "almost fell asleep" (for analyses, see Supplementary Table 1). No resting-state data had to be excluded due to sleep.

MRI data were not available due to technical problems from $n = 0/3/3$ participants on days 1/2/3.

**Preprocessing of fMRI data.** Only task and resting-state fMRI data from day 2, relevant to address our hypotheses, were analyzed and are reported here. fMRI data was preprocessed and analyzed using statistical parametric mapping (SPM12, Wellcome Trust Center for Neuroimaging, London, UK, http://www.fil.ion.ucl.ac.uk/) running on Matlab 2015b (MathWorks®, Natick, Massachusetts, USA). The first 5 volumes of each scan were discarded due to equilibrium effects. Preprocessing includes realignment and co-registration of the mean functional image to the T1 weighted anatomical image. Subsequently, the T1 weighted anatomical image was segmented and normalized to Montreal Neurological Institute (MNI) space based on SPM's tissue probability maps. Normalization of the functional images was achieved by applying the resulting deformation fields to the realigned and co-registered functional images. Lastly, functional data was smoothed using a 6 mm full-

width-at-half-maximum Gaussian smoothing kernel. Data of participants was excluded when movement peaks exceeded more than 3 mm or 2° (task data, $n = 7$, resting-state data: $n = 6$ further participants).

**Data analysis**

**Statistical analysis of behavioral and psychophysiological data.** For statistical analysis of SCR and US-expectancy ratings, we conducted repeated measures ANOVA with stimulus (CS + /CS−) as within- and drug (placebo/L-DOPA) as between-subject factor. Following our previous studies[16,18,19] and preregistration, we tested whether fear was successfully acquired based on CRs averaged across the last 20% of trials during fear conditioning (i.e., last 2 trials). Start-fear at extinction on day 2 was assessed using the average first 20% of trials during extinction (i.e., first 3 trials). End-fear of extinction was determined across the last 20% of trials (i.e., last 3 trials) of the extinction session on day 2. The effect of L-DOPA on retrieval was tested using CRs averaged across all trials of either the spontaneous recovery or the renewal test on day 3. For each statistical test, it was ensured that assumptions were met. All implemented t-tests were two-tailed. For all models that included trait and state sAA, we examined the model assumptions such as normality of residuals, homoscedasticity, and linearity to ensure their validity. All statistical analyses were implemented in R version 4.1.2 (2021-11-01). Analyses have been conducted using the following packages: ez (ANOVAs), car, MASS (regression analyses and diagnostics), pequod, emmeans (simple slope analyses), rstatix, BayesFactor (statistical tests), irr (ICC), lmerTest (linear mixed effects models), coefplot, ggeffects, sjPlot stats, DescTools (diagnostics).

**Multivariate fMRI analysis.** Investigating potential reactivations of extinction specific MVPs in the vmPFC, we analyzed the extinction task data (day 2) using a model including one regressor for CS+ and CS− onsets, respectively, US-expectancy ratings, and context on/-offset. Furthermore, the model included one regressor for the first five CS+ offsets, where omission of the US was unexpected, and one for the first five CS- offsets, where US omission was expected, as well as one regressor each for the remaining ten CS+ and CS- offsets. All regressors were delta-functions convolved with the hemodynamic response function (HRF). The MVP evoked by the first five US omissions at CS+ offset in the vmPFC was extracted from the resulting beta-map in the vmPFC region of interest (ROI; see below). Resting-state data was analyzed in accordance with a previous study examining memory reactivation[83] (see also ref. [16]), i.e., general linear models (GLMs) for each day 2 resting-state scan (pre-, -10, 45, and 90 min post-extinction) included delta-function regressors for each volume (TR: 1 s), thereby accounting for potential reactivations which may have occurred during any point of the resting-state scan. No high-pass filtering was applied in the resting-state models and AR(1) auto-correlation correction was employed. MVPs in the pre-defined ROI during the resting state were extracted from the resulting beta-image series (TR = 1 s, i.e., $480 - 5 = 475$ beta images).

Subsequently, we correlated (Pearson correlation coefficient) the pattern evoked by the first five US omissions at CS+ offset during extinction with the resulting 475 patterns of all four resting-state scans and Fisher Z- transformed the correlation coefficients. The 475 correlations of the US omission pattern with the resting-state pattern recorded before extinction learning was employed to create a baseline distribution. The mean and standard deviation of this baseline distribution was used to transform each correlation between the template and the resting-state patterns into a Z- score ($Z_i = (r_i - \mu_i)/\sigma$). Correlations with a Z-score exceeding a value of 2 ($Z > 2 \approx p < 0.05$) were counted as potential reactivations of the CS+ offset-related vmPFC pattern. Reactivations were summed per participant and resting-state scan. Subsequently, multiple linear regression analyses with number of CS+ offset-related vmPFC reactivations at baseline, -10, 45, and 90 min after extinction as predictors and average differential (CS + > CS−) SCR

during either the spontaneous recovery or the renewal test as dependent variables were performed separately.

**Regions of interest (ROIs).** We focused our analysis on the vmPFC based on previous work[16]. Control regions involved in fear and extinction learning were included comprising of: anterior cingulate cortex (ACC), superior frontal gyrus (SFG), left and right insula, left and right amygdala, and left and right hippocampus (see Supplementary Figs. 2 and 3 for results). All ROIs were extracted from the Harvard-Oxford Atlas and thresholded at 50%-tissue probability by previous work[16].

### Reporting summary
Further information on research design is available in the Nature Portfolio Reporting Summary linked to this article.

## Data availability
The sAA, sCORT, US expectancy, questionnaire and rating raw data generated in this study have been deposited in Zenodo under https://doi.org/10.5281/zenodo.8353754. The raw MRI data are not available and cannot be shared upon request due to data privacy laws. Specifically, participants did not consent to these data being shared. Anonymization as understood by the EU General Data Protection Regulation (GDPR) is not possible because binding internal regulations of the University Medical Center Mainz require that all MRI raw data acquired at the hospital are stored alongside their names and other identifying data on a central hospital server for the duration of 30 years. As a consequence, there is a risk that individuals could be identified by cross-referencing shared data with stored information. The processed MRI data (vmPFC reactivations) and derived SCR data are available at the link provided above.

## Code availability
The custom-made analysis script for scoring SCR running on Matlab 2015b and an R script running on R version 4.3.0 testing the three main hypotheses and the secondary research questions can be accessed via https://doi.org/10.5281/zenodo.8353754.

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

## Acknowledgements

We thank K. Yuen for support with data collection and analysis and J. Meier for support with data collection. This work was funded by the Deutsche Forschungsgemeinschaft (DFG), SFB1193, subproject C01 to R.K.

## Author contributions

E.A., C.-P.H., A.M.V.G., O.T., and R.K. designed the experiment; E.A. and C.-P.H. collected the data; E.A., C.-P.H., and B.M. analyzed the data; and E.A. and R.K. wrote the paper.

## Funding

## Competing interests

The authors declare no competing interests.
