## [Peer Review File · Nature Communications]

Trait salivary alpha-amylase activity levels in males define the conditions for facilitation by L-DOPA of extinction consolidationREVIEWER COMMENTS

Reviewer #1 (Remarks to the Author):

The study by Andres et al., presents a preregistered replication of the Gerlicher et al., 2018 paper, showing that spontaneous reactivation of neural activity patterns related to CS+ offset in post-extinction rest predicts later extinction recall, and that L-DOPA enhances this effect. Andres et al. find an association between vmPFC pattern activation and later extinction recall but identify boundary conditions for the effects of L-DOPA. The original study by Gerlicher et al. proposed a potential mechanism underlying the (clinically relevant) finding that L-DOPA can promote extinction memory consolidation - given the novelty of those findings and the general lack of replicability and power issues in human neuroimaging studies, replication is much needed. The fact that the same lab attempts such a difficult replication, instead of leaving this to other researchers and moving on to the next hot topic, is laudable. The paper is generally well-written, and the results are thought-provoking. I have carefully read the study preregistration from 2018 and compared it to the final manuscript. I have a few questions, mainly to confirm that data that would complicate the present story have not been omitted, and to judge the specificity of the vmPFC effects.

1 The paper presents the relation between neural pattern reactivation in the 90-min block and subsequent extinction memory retrieval as evidence for one of the three hypotheses, while the study by Gerlicher et al., finds the effect in the 45-min block, and this is also where the preregistration predicted it. While this is mentioned in the Introduction and Results, it is not mentioned in the Abstract and Discussion and feels a bit brushed away by statements that results 'confirm' (abstract) or 'firmly establish' (page 9, 17) the role for spontaneous post-extinction reactivations in retrieval of extinction memory. The consolidation window may be broad, but for an exact replication this level of variability seems surprising: one may just as well argue that the current paper does not replicate previous findings. Therefore, the conclusion needs toning down (and multiple comparisons correction for the number of tested intervals would be neat).

2. Relatedly, it's not explained why the 45-min window is abandoned for all analyses as soon as a significant result is found for the vmPFC in the 90-min time point. For example, L-DOPA could still have a significant effect on activity patterns in the 45-min scan when controlling for sAA, even if there is no main effect. This is relevant especially because dopamine is thought to peak in this time window. In addition, the anatomical specificity of the effects is only tested for the 90-minute scan, showing that the vmPFC uniquely shows the reactivation- retrieval relation, but what about other areas at 45 min? The special role of the vmPFC would likely be challenged if other regions show an effect in the 45-min window, so it would be informative to present the data in the Supplement (as is now done for the 90 minute scan in S Fig 2. For my understanding, why does the Figure legend report a weaker effect for the vmPFC here than reported on page 9? Is it because this analysis shows the unique variance rather than total variance explained?)

3. The secondary analysis on sAA should - according to the preregistration - be conducted using the individual peak increase in salivary alpha-amylase and cortisol (peak-baseline) as predictor. Instead, in the manuscript the authors take the Pre-extinction levels as a predictor for everyone, regardless of whether this was the peak for a particular individual. Presumably, the reason for this is that on average, sAA decreased in the entire group (a surprising finding?), so for many the peak is the baseline, but there may still be some people with a different peak, which would change the analyses presented in Figure 5 and 6.

4. Not preregistered, but something that was done in Gerlicher et al., 2018 and that would strengthen the idea that there really is someone unique to the US omission pattern (the idea that PE plays a role) is if the authors can show that the CS- offset pattern reactivation at 45 or 90 minutes does indeed not predict extinction memory recall.

5. Relatedly, the results in Figure 4 don't show CS+ pattern 'reactivation' during pre-extinction

baseline, though this was assessed according to the preregistered data analysis plan. Adding these data helps comparing the results to those reported in Gerlicher 2018, Figure 2e, and again, strengthen the notion that this some form of adaptive replay of PE signals.

6. Univariate analyses and analyses with MVP as predictor for differential BOLD responses and task-modulated functional connectivity on day 3 are preregistered but not reported. Please clarify why not.

Reviewer #2 (Remarks to the Author):

This manuscript describes a replication and extension study of the pro-extinction impact of L-DOPA, the potentially mediating role of vmPFC reactivations, and the potentially moderating role of arousal during extinction. This is a well-designed and well-conceived study whose pre-registration is a strength. The authors have done well with extensively reporting important methodological details and analyses, and differentiating between pre-registered vs post-hoc analyses. The results are in line with the hypothesis that arousal during extinction serves as a boundary condition for L-DOPA to enhance extinction consolidation. This study has the potential to make a major contribution to the literature. I have a few comments to improve the manuscript and its conclusions.

One important consideration is whether the arousal markers defined during pre-extinction are unique to an individual's state during extinction, or whether they reflect a trait-like tendency / propensity for arousal. This is important because it impacts the interpretation of relationships between pre-extinction arousal and other indices (e.g., vmPFC reactivation, spontaneous recovery, etc). The competing hypotheses here are that a) something about the state (high arousal) predicts differences in the outcome, or b) something about the person (a person who tends to be highly aroused) predicts differences in the outcome. The authors seem to be leaning heavily in to the former without much consideration of the latter. Because the authors collected saliva on other days, they can actually adjudicate between these hypotheses. A day (day1 vs day2 vs day3) x sAA mixed model predicting an outcome variable (eg spontaneous recovery) should yield a day x sAA interaction under the former hypothesis, such that sAA uniquely at pre-extinction predicts outcome. Under the latter hypothesis, there would be a main effect of sAA such that an individual's sAA levels in general predict poorer outcomes. Further, it would be helpful for interpretation to report the correlations between day1, day2, and day3 sAA levels in order to further document how unique day2 levels are to day 2. I realize these are not a pre-registered analyses, but I believe they are essential to interpret the results.

The connectome predictive modeling analyses, which nicely link pre-extinction sAA to a LC network FC and not to whole-brain or dopamine network FC, could similarly be replicated at pre-day 1 and pre-day3 resting state scans. That is, if sAA is indeed a marker of this network FC, it should be observed at other timepoints too.

Similarly, the authors collected the STAI-S at multiple timepoints, was this related to sAA? If not, it would have implications for clinical translation. That is, anxiety is heightened during exposure. So if it is sAA, but not anxiety, that is required for L-DOPA to facilitate consolidation, then it may not necessarily be good news for clinicians if there is a clear dissociation between self-reported anxiety (which clinicians routinely measure) and sAA (which clinicians do not routinely or easily obtain).

Finally, I did not see consideration of the lack of an effect on threat expectancies in the discussion. While skin conductance provides an ostensibly objective measure, and were the focus of the original paper, it is only a measure of arousal, which is only one facet of the larger construct of negative emotional responding to the conditioned cue. Since there was no effect on threat expectancies, which again are likely as relevant, if not more relevant, to clinical practice, at least

mention of this the discussion is warranted, possibly along the lines of the non-unitary nature of fear responding and that different outcomes might reflect different underlying processes.

Reviewer #3 (Remarks to the Author):

The present study is a replication of previous findings on fear extinction. It could be confirmed that neuronal activation patterns in the ventromedial prefrontal cortex during extinction can predict the retrieval of extinction memory 24 hours later. In contrast, an effect of levodopa on extinction memory has not been confirmed.

In general, this is a very elaborate and in large parts very well controlled study with sufficient statistical power. Replication study guidelines were followed, and all unregistered analyses were clearly labelled. The results of the replication study are of great interest to the field.

In addition, the relationship between salivary alpha-amylase activity and locus coeruleus connectivity and its importance for extinction retrieval is investigated. The results show that levodopa can enhance extinction consolidation during high arousal. The relevance of the results is discussed for exposure therapy sessions, which are typically associated with high arousal.

The latter results in particular are extremely interesting and may make an important contribution to our understanding of extinction retrieval.

However, from a methodological perspective, they are very difficult to assess. In particular, the manuscript has weaknesses in the sampling and analysis of salivary alpha-amylase and in the measurement of locus coeruleus connectivity patterns.

specific comments:

1) Salivary alpha-amylase (sAA).

The key to reliable interpretation of salivary alpha-amylase activity is sampling and analysis. Both are not well described in the manuscript, and some standards may not have been followed.

Therefore, it is important that essential information is added first in order to interpret the data appropriately.

- It is unclear whether stimulation of sAA occurred. For example, chewing can stimulate sAA secretion. Accordingly, it is important to know the exact instructions that subjects received during saliva collection. Was there any standardization of saliva collection? How long were the collectors kept in the mouth? Were subjects instructed to chew on the collectors or did they simply place them in the mouth? Also, since secretion of sAA varies from salivary gland to salivary gland, it is important to know where the collector was placed when it was not moved in the mouth. Please fill in the information as accurately as possible.

- Cotton swabs are rather unsuitable for SAA analyses. They cannot be completely centrifuged and the amylase starts to decompose the absorbent cotton, i.e. the enzyme activity increases (therefore synthetic collectors are recommended). In any case, you should specify exactly which collectors were used (e.g. "Sallivettes white cap" with cotton swabs, "Manufacturer") and how much time elapsed between sample collection and freezing and thawing and analysis. Please fill in the information as accurately as possible.

- To my knowledge, the sAA tests do not determine the sAA concentration in saliva, but the metabolic activity in units per ml. SAA concentration is a combined effect of flow rate and protein secretion. This should be corrected.

- Finally, the direct relationship between sAA and noradrenergic/sympathetic activity is not well established. Reactivity measures such as the increase in sAA are useful for a cautious estimate of noradrenergic or sympathetic activity (e.g., Dietzen et al., 2014; <http://dx.doi.org/10.1016/j.biopsycho.2014.08.001>). However, in the present study, no reactivity measures were used, but the sAA values before extinction. These values are very difficult to interpret, which is a clear limitation.

- The Interpretation of amylase activity as a direct measure of physiological excitation is also difficult unless it is caused by a unique condition or can be confirmed by behavioural data.

The following paper provides a good overview of the points mentioned above:

- Reference: Bosch, J. A., Veerman, E. C. I., de Geus, E. J., & Proctor, G. B. (2011). α -Amylase as a reliable and convenient measure of sympathetic activity: Don't start salivating just yet! *Psychoneuroendocrinology*, 36(4), 449–453. <https://doi.org/10.1016/j.psyneuen.2010.12.019>

2) Use of the locus coeruleus as a seed region for connectivity measurements

- The LC is very small and surrounded by large blood vessels and ventricles. Functional measurements of the structure have high requirements for adequate mapping of the structure. A 2009 paper by Keren is cited as a reference for measuring the LC. However, a debate arose about the reliable measurement of the activity of the LC and similar small structures. It has been suggested that many results may be erroneous because mapping of the LC is very difficult, and pulsation of nearby blood vessels and motion artifacts due to proximity to the ventricles can greatly affect measurements.

Reference: Astafiev, S. V., Snyder, A. Z., Shulman, G. L., & Corbetta, M. (2010). Comment on "Modafinil Shifts Human Locus Coeruleus to Low-Tonic, High-Phasic Activity During Functional MRI" and "Homeostatic Sleep Pressure and Responses to Sustained Attention in the Suprachiasmatic Area". *Science*, 328(5976), 309–309.

- In an eLetter published along with the publication, <https://www.science.org/doi/10.1126/science.1177200>

the editors and other experts in the field ultimately recommend that future studies ...

- Perform explicit brainstem normalization
- Use standardized LC masks as ROIs
- And should perform cardiac gating to control pulsation

I am aware that the above points are not fully feasible in such an elaborate study. However, it would increase confidence in the data if the connectivity maps were shown within the LC masks. For this purpose, the regions whose LC connectivity directly correlates with sAA (vSTR_L, vSTR_R, dmPFC) could be used as seed regions in control analyses. Subsequently, it can be visualized whether their connectivity patterns in the brainstem lie within standardized masks of the LC. If not already done, brainstem normalization would be required.

Example of LC masks related to the above eLetter could be found here: www.eckertlab.org/lc

Please also indicate if the left and right LC time series were averaged for the analysis in the initial analysis or if the left and right LC time series were specified individually in the model.

Other points:

3) Figure 7.

If I understand correctly, ROI-to-ROI whole-brain analyses were performed over 400 cortical and subcortical regions, not seed-to-voxel analyses. If the ventral striatum was one of the ROIs, why is there a light blue cluster in the left posterior putamen in Figure 7? Please explain and describe the results in more detail.

What is the difference between dACC 1 and dACC2? Please explain.

4) Replication:

- Please briefly state how the technical parameters / scanner / electroshock ect. were comparable to the initial study.

5) Statistical trends:

- Trend wise association are non-significant results and should be presented as such.

6) Stress

-In the study, stress was not explicitly measured (e.g., via a questionnaire). Levels of sAA and cortisol varied depending on many factors and could not be interpreted as independent markers of stress. No measures of sAA reactivity were reported, and it was not tested whether there was an increase in cortisol above the normal pulse rate (responder analysis). This is also not possible with

the two saliva samples before and after the respective measurements. I therefore advise against using the term stress and arguing with more extensive stress regulation patterns. Statements like: „ sAA and sCORT originate from two different stress systems, the first being faster and more sensitive in responding to stressors, while the second responds more slowly and to more stressful tasks. Our current results suggest that acute arousal system activation (related to sAA) might have stronger direct effects on extinction than HPA system activation (related to sCORT) and show more important interactions with dopaminergic transmission.“

suggest that there was a solid activation of both stress systems. However, this was not tested and so it is unclear whether the sAA and cort values reflect a stress response or the normal circadian course.

7) The journal explicitly asks the reviewers to pay attention to compliance with the SAGER guidelines and to check whether the analyses are in accordance with the preregistration.

- In the present study, only men were examined, which is methodologically justified.
- However, the SAGER guidelines require that it should be made clear in the title and abstract that only one sex is involved. The title and abstract should be adjusted accordingly.

- The discussion should include a point in which it is made clear that the results apply to only one sex.

-Based on my review, the analyses do not differ from the preregistered analyses. Some analyses have been omitted due to technical issues. Additional analyses have been clearly identified.

Point-by-point reply

“Trait salivary alpha-amylase activity levels define the conditions for facilitation by L-DOPA of extinction consolidation”

Andres et al.

Introductory remarks to all reviewers

We greatly appreciate the time and effort taken by all reviewers in assessing our manuscript. We thank the reviewers for their constructive questions and intriguing suggestions. These have importantly shaped the paper.

To highlight the most impactful comments, reviewer 1 has pointed out that it is important to conduct our analyses of the influence of the salivary alpha-amylase (sAA) in response to the experimental manipulation on the experimental day 2 (being placed in an MRI scanner and experiencing fear extinction training) on extinction and the effect of L-DOPA treatment with individual rather than group-level peaks, which we have implemented.

In response to reviewer 2's valuable input, we would like to express our gratitude for their bringing to our attention the importance of analyzing sAA activity levels not only as a measure of the acute reactivity of the sympathetic nervous system (SNS) on day 2 but also as a trait marker that expresses stable individual differences in SNS activity. Guided by the reviewer and a literature that was previously unknown to us, we indeed observed high correlations between the baseline values of sAA on the three experimental days and found a high test-retest reliability (ICC). On this basis, we now operationalize 'trait sAA' as the average of these three baseline sAA measures. This new approach has unveiled a strong association between high trait sAA and impaired extinction learning and memory, as well as a notable interaction between trait sAA and L-DOPA treatment, indicating that L-DOPA is particularly effective in improving extinction memory consolidation in individuals with elevated trait sAA.

Interestingly, the impact of state sAA (peak activity on the extinction day) appears relatively less important, and the previously reported state sAA by L-DOPA interaction is no longer present in the unified model combining trait and state sAA indices. This finding is particularly intriguing considering the high reliability and heritability of trait sAA reported in the literature, as well as the comparatively weak reliability of state sAA as an index of acute arousal or stress. Moreover, given that trait sAA has been shown to be a reliable marker for the basal activity of the SNS and has previously been linked to cognitive function, we now possess a stable and physiologically meaningful predictor for the success of extinction retrieval and the efficacy of L-DOPA. This discovery has significant clinical implications, potentially paving the way for a new approach to precision medicine. Specifically, L-DOPA as an enhancer of exposure therapy could be selectively administered to individuals with high trait sAA. We emphasize that this new finding is based on a non-preregistered analysis and requires replication.

Reviewer 3 has provided important methodological hints regarding the analysis of the functional connectivity (FC) of the locus coeruleus (LC) network in the brain. Reanalyzing the data following their suggestions showed that we can no longer claim an association of LC FC and state sAA (nor trait sAA). We have therefore removed these non-preregistered analyses from the paper. We thank the reviewer for protecting us against reporting false positive findings.

The described new findings have led to substantial modifications in the manuscript, allowing us to provide a more comprehensive and impactful analysis of our research.

We have attached one version of the manuscript with all changes tracked and one cleaned version with all changes accepted.

Please find below our detailed point-by-point responses.

REVIEWER COMMENTS

Reviewer #1 (Remarks to the Author):

REVIEWER: The study by Andres et al., presents a preregistered replication of the Gerlicher et al., 2018 paper, showing that spontaneous reactivation of neural activity patterns related to CS+ offset in post-extinction rest predicts later extinction recall, and that L-DOPA enhances this effect. Andres et al. find an association between vmPFC pattern activation and later extinction recall but identify boundary conditions for the effects of L-DOPA. The original study by Gerlicher et al. proposed a potential mechanism underlying the (clinically relevant) finding that L-DOPA can promote extinction memory consolidation - given the novelty of those findings and the general lack of replicability and power issues in human neuroimaging studies, replication is much needed. The fact that the same lab attempts such a difficult replication, instead of leaving this to other researchers and moving on to the next hot topic, is laudable. The paper is generally well-written, and the results are thought-provoking. I have carefully read the study preregistration from 2018 and compared it to the final manuscript. I have a few questions, mainly to confirm that data that would complicate the present story have not been omitted, and to judge the specificity of the vmPFC effects.

1 The paper presents the relation between neural pattern reactivation in the 90-min block and subsequent extinction memory retrieval as evidence for one of the three hypotheses, while the study by Gerlicher et al., finds the effect in the 45-min block, and this is also where the preregistration predicted it. While this is mentioned in the Introduction and Results, it is not mentioned in the Abstract and Discussion and feels a bit brushed away by statements that results 'confirm' (abstract) or 'firmly establish' (page 9, 17) the role for spontaneous post-extinction reactivations in retrieval of extinction memory. The consolidation window may be broad, but for an exact replication this level of variability seems surprising: one may just as well argue that the current paper does not replicate previous findings. Therefore, the conclusion needs toning down (and multiple comparisons correction for the number of tested intervals would be neat).

AUTHORS: We thank the reviewer for their appreciation of our work and their valuable comments. Firstly, let us reiterate our introductory remarks above that the manuscript has undergone significant changes in response to the reviewer comments, particularly with the inclusion of the new finding that L-DOPA effectiveness in enhancing extinction memory consolidation is predicted by trait sAA.

Regarding the issue of variability in the replication of previous findings, we understand the concern raised by the reviewer. While the study by Gerlicher et al. found the effect in the 45-min block, our preregistration explicitly allowed us to test reactivations at multiple time points, considering the potentially broad time window of L-DOPA's action on consolidation (see also L116 of the new manuscript version). Moreover, as L-DOPA is orally administered, there can be pharmacokinetic variations that may influence the peak plasma concentration and timing of the observed effects. We appreciate the reviewer highlighting the need to acknowledge this variability and have addressed it in the revised manuscript. We have omitted "firmly" from the sentence "these results establish an important role for spontaneous post-extinction reactivations of an extinction-related activity pattern in the vmPFC in the consolidation of long-term extinction memories" in L244 in the Results. We have added "though not at 45 min as in the discovery study" on L375 in the Discussion. We explicitly discuss the question of variability starting L425 and have added to this section: "It is also worth noting that oral administration of a drug can lead to substantial pharmacokinetic variability." Finally, we have incorporated the Bonferroni correction as suggested by the reviewer and now report that

the vmPFC reactivations at 90 minutes remained significant after applying the correction L232. We do maintain that our study confirms a predictive relationship between post-extinction vmPFC reactivations and extinction retrieval.

REVIEWER: 2. Relatedly, it's not explained why the 45-min window is abandoned for all analyses as soon as a significant result is found for the vmPFC in the 90-min time point. For example, L-DOPA could still have a significant effect on activity patterns in the 45-min scan when controlling for sAA, even if there is no main effect. This is relevant especially because dopamine is thought to peak in this time window.

AUTHORS: We agree with the reviewer's suggestion that L-DOPA could still have a significant effect on activity patterns in the 45-minute scan when controlling for sAA, even if there is no effect in the prior analysis not controlling for sAA. We therefore now also analyzed the 45 min time point in the analyses controlling for sAA, however, finding no effects. See L351.

REVIEWER: In addition, the anatomical specificity of the effects is only tested for the 90-minute scan, showing that the vmPFC uniquely shows the reactivation- retrieval relation, but what about other areas at 45 min? The special role of the vmPFC would likely be challenged if other regions show an effect in the 45-min window, so it would be informative to present the data in the Supplement (as is now done for the 90 minute scan in S Fig 2. For my understanding, why does the Figure legend report a weaker effect for the vmPFC here than reported on page 9? Is it because this analysis shows the unique variance rather than total variance explained?)

AUTHORS: We now also conducted analyses for reactivations in all brain regions at the 45-minute time point. Supplementary Figure 3 shows that no evidence for reactivations being predictive of extinction retrieval in the vmPFC or any other of the ROIs used in Gerlicher et al. (2018) was found at 45 minutes in the analyses not controlling for sAA (L240). This strengthens the case for the special role of the vmPFC in the reactivation-retrieval relationship.

Regarding the discrepancy in reported effects between the figure legend and page 9 of the old manuscript, we apologize for any confusion caused. The difference in reported effects is due to the use of different models for analysis. In the model presented on page 9, we included predictors for the three different time points only in the vmPFC, whereas the model in the supplementary figure legend focused on a single time point but included predictors for different ROIs. By including more predictors in the model with other ROIs, we account for shared variance among different brain regions. This additional complexity in the model can influence the statistical significance and magnitude of individual predictors. We have now added in the Supplementary Figure 2 legend: When including all regions' number of reactivations, together with the number of reactivations in the vmPFC, into a regression model, the number of CS+ offset-related pattern reactivations in any other region except for the vmPFC did not predict CRs at test (spontaneous recovery in context B) (multiple linear regression: vmPFC: $\beta = -.07$, $p = .036$; all other $p > .11$; $n = 46$).

REVIEWER: 3. The secondary analysis on sAA should – according to the preregistration – be conducted using the individual peak increase in salivary alpha-amylase and cortisol (peak-baseline) as predictor. Instead, in the manuscript the authors take the Pre-extinction levels as a predictor for everyone, regardless of whether this was the peak for a particular individual. Presumably, the reason for this is that on average, sAA decreased in the entire group (a surprising finding?), so for many the peak is the baseline, but there may still be some people with a different peak, which would change the analyses presented in Figure 5 and 6.

AUTHORS: Firstly, please note that we now use the term 'state sAA' throughout the manuscript for the individual reaction in sAA activity levels to the experimental manipulation (scanning, extinction) on day 2, as a result of the intervention of reviewer 2 (see also introductory remarks). We acknowledge that our operationalization of state sAA in the old manuscript version was a deviation from the preregistered approach and thank the reviewer for pointing this out. The decision to use pre-extinction sAA as predictor of extinction retrieval in all participants was primarily motivated by the unexpected finding of a decrease in sAA over time on average for the entire group on day 2 (now

more extensively described starting L262 of the new manuscript version and discussed starting L445). We have now corrected this and incorporated each individual's peak as predictor, better reflecting the state sAA, in our new common model alongside trait sAA (L275 ff.). Here, approximately 30% of participants exhibited their sAA peak immediately after extinction rather than before. This operationalization is still a slight deviation from the preregistration, where we announced we would use the peak minus baseline difference. This is, however, not possible for the majority of individuals, who show their peak at baseline. The deviation from preregistration is explicitly mentioned in L289 ff.

REVIEWER: 4. Not preregistered, but something that was done in Gerlicher et al., 2018 and that would strengthen the idea that there really is someone unique to the US omission pattern (the idea that PE plays a role) is if the authors can show that the CS- offset pattern reactivation at 45 or 90 minutes does indeed not predict extinction memory recall.

AUTHORS: At the request of the reviewer, we have conducted additional analyses to investigate the uniqueness of the US omission pattern in relation to extinction memory retrieval. We have included the statistics for the CS- offset related pattern in the vmPFC in the manuscript on L241, showing that this is not predictive of extinction memory retrieval.

REVIEWER: 5. Relatedly, the results in Figure 4 don't show CS+ pattern 'reactivation' during pre-extinction baseline, though this was assessed according to the preregistered data analysis plan. Adding these data helps comparing the results to those reported in Gerlicher 2018, Figure 2e, and again, strengthen the notion that this some form of adaptive replay of PE signals.

AUTHORS: Please see the adjusted Figure 4.

REVIEWER: 6. Univariate analyses and analyses with MVP as predictor for differential BOLD responses and task-modulated functional connectivity on day 3 are preregistered but not reported. Please clarify why not.

AUTHORS: The reviewer is right, and we apologize for omitting this in the old manuscript. We indeed pre-registered these analyses, however, did not include them in our primary or secondary research hypotheses in the preregistration. We now state L247: "The number of vmPFC reactivations at any of the post-extinction time points did not detectably predict differential BOLD responses (contrast CS+ > CS-) on day 3. Accordingly, the preregistered analysis of the effects of vmPFC reactivations on task-modulated functional connectivity (contrast CS+ > CS-) on this day could not be conducted."

We thank the reviewer for their careful reading of preregistration and manuscript, which has allowed us to improve the text significantly.

Reviewer #2 (Remarks to the Author):

REVIEWER: This manuscript describes a replication and extension study of the pro-extinction impact of L-DOPA, the potentially mediating role of vmPFC reactivations, and the potentially moderating role of arousal during extinction. This is a well-designed and well-conceived study whose pre-registration is a strength. The authors have done well with extensively reporting important methodological details and analyses, and differentiating between pre-registered vs post-hoc analyses. The results are in line with the hypothesis that arousal during extinction serves as a boundary condition for L-DOPA to enhance extinction consolidation. This study has the potential to make a major contribution to the literature. I have a few comments to improve the manuscripts and its conclusions.

One important consideration is whether the arousal markers defined during pre-extinction are unique to an individual's state during extinction, or whether they reflect a trait-like tendency / propensity for arousal. This is important because it impacts the interpretation of relationships between pre-extinction arousal and other indices (e.g., vmPFC reactivation, spontaneous recovery, etc). The competing hypotheses here are that a) something about the state (high arousal) predicts

differences in the outcome, or b) something about the person (a person who tends to be highly aroused) predicts differences in the outcome. The authors seems to be leaning heavily in to the former without much consideration of the latter. because the authors collected saliva on other days, they can actually adjudicate between these hypotheses. A day (day1 vs day2 vs day3) x sAA mixed model predicting an outcome variable (eg spontaneous recovery) should yield a day x sAA interaction under the former hypothesis, such that sAA uniquely at pre-extinction predicts outcome. Under the latter hypothesis, there would be a main effect of sAA such that an individual's sAA levels in general predict poorer outcomes. Further, it would be helpful for interpretation to report the correlations between day1, day2, and day3 sAA levels in order to further document how unique day2 levels are to day 2. I realize these are not a pre-registered analyses, but I believe they are essential to interpret the results.

AUTHORS: We sincerely appreciate the reviewer's input, which has allowed us to make what we think is an important new discovery. We apologize that we were not aware at the time of the initial analysis of our data of the literature on sAA as a trait marker. In our new analyses inspired by the reviewer, the distinction between state and trait components of sAA has indeed turned out to be crucial for a comprehensive understanding of the implications of sAA in relation to various outcomes, such as extinction retrieval (inverse spontaneous recovery) and post-extinction vmPFC reactivations.

The correlations between baseline sAA values (meaning, on day 2: pre-extinction) across days were strong and highly significant (day 1-2: $R=.46$; day 1-3: $R=.61$; day 2-3: $R=.60$). See L275 ff. of the new manuscript version. Additionally, test-retest reliability was high, with an ICC of 0.701 ($p<0.0001$). These observations strongly support the reviewer's idea of a trait-like or personality-like effect in our sAA data. This notion was also in line with the results of our literature search, which we report L159 and L453 ff. and which also revealed substantial heritability and the possibility to use average sAA measures in a state of rest, or baseline, from as little as three days to obtain reliable trait markers.

Considering these findings, we have made the necessary adjustments in our approach. Specifically, we have established a new model incorporating the trait sAA value (average of baseline values) and the deviation of the individual's peak from their average on extinction day 2, which is a non-collinear term that can be taken to reflect the current state of sAA activity around extinction on that day (formerly addressed to as the sAA reaction to the experimental manipulation on that day). These variables, along with group and the group by trait and by state interactions, are now included in all prediction analyses. As a result of this refined approach, we have discovered a strong negative trait sAA effect on extinction retrieval and a meaningful trait by group interaction, in the form of enhanced efficacy of L-DOPA at high trait sAA (L296 ff.). Trait (but not state) sAA also predicted poorer extinction learning (L315 ff.). In the analysis of vmPFC reactivations (L331 ff.), there were trend-level effects.

We recognize the clinical relevance of these findings and their potential implications for treatment approaches. It is our hope that these novel insights will contribute to a new precision medicine perspective, where individuals with high trait sAA (which can be reliably and relatively uncostly determined) can benefit from exposure therapy augmentation with L-DOPA.

Effects of state sAA were comparatively less important, although state sAA did impact vmPFC reactivations (L331 ff.).

REVIEWER: The connectome predictive modeling analyses, which nicely link pre-extinction sAA to a LC network FC and not to whole-brain or dopamine network FC, could similarly be replicated at pre-day 1 and pre-day3 resting state scans. That is, if sAA is indeed a marker of this network FC, it should be observed at other timepoints too.

AUTHORS: We appreciate the reviewer's suggestion to replicate the connectome predictive modeling analyses at pre-day 1 and pre-day 3 resting state scans to further investigate the relationship between pre-extinction (baseline) sAA and LC network functional connectivity. Upon conducting additional analyses, we found that the relationship between the LC network and sAA was not observed on the other days. Furthermore, we examined the test-retest reliability of the baseline

LC network measurements and found that they were much less reliable (ICC =0-0.3) compared to the baseline sAA measurements (ICC =0.7). Additionally, we concatenated the baseline resting state scans to obtain the mean baseline LC network used to predict the trait-like (averaged) amylase levels. The mean baseline LC network did neither predict amylase levels on day 1-3 nor trait-like amylase. Given these findings, it appears that the LC network measurement is less suitable as an individual-differences marker. We have therefore made the decision to exclude the LC network analysis from the manuscript.

REVIEWER: Similarly, the authors collected the STAI-S at multiple timepoints, was this related to sAA? If not, it would have implications for clinical translation. That is, anxiety is heightened during exposure. So if it is sAA, but not anxiety, that is required for L-DOPA to facilitate consolidation, then it may not necessarily be good news for clinicians if there is a clear dissociation between self-reported anxiety (which clinicians routinely measure) and sAA (which clinicians do not routinely or easily obtain).

AUTHORS: There were no significant correlations between the STAI-S scores and trait or state sAA in our data. This suggests that trait sAA, rather than self-reported anxiety, is the relevant factor for the facilitation of consolidation with L-DOPA. We do think that determining trait sAA is feasible and can be incorporated into clinical routine at reasonable cost, especially considering the potential benefit of guiding pharmacological augmentation.

REVIEWER: Finally, I did not see consideration of the lack of an effect on threat expectancies in the discussion. While skin conductance provides an ostensibly objective measure, and were the focus of the original paper, it is only a measure of arousal, which is only one facet of the larger construct of negative emotional responding to the conditioned cue. Since there was no effect on threat expectancies, which again are likely as relevant, if not more relevant, to clinical practice, at least mention of this the discussion is warranted, possibly along the lines of the non-unitary nature of fear responding and that different outcomes might reflect different underlying processes.

AUTHORS: While our focus in this as well in all our previous studies on L-DOPA in extinction was on skin conductance as an implicit and objectively measurable conditioned response, we acknowledge the significance of considering threat expectancies, especially in clinical practice. In the discussion, L524 ff., we write: "Beyond the role of salivary measures, it should be noted that all reported effects on extinction retrieval in this study were observed in skin conductance responses as our main index of conditioned responding. There were no predictive relationships of vmPFC reactivations, sAA, or drug treatment on US expectancy ratings. SCRs are implicit and objectively measurable, but expectancy or other ratings have the practical advantage that they can be easily collected, in particular also in clinical settings. One explanation for the absence of effects on ratings may lie in fear responses being carried by dissociable systems (e.g., Soeter and Kindt, 2012), not all of which may be affected by our manipulation."

Reviewer #3 (Remarks to the Author):

REVIEWER: The present study is a replication of previous findings on fear extinction. It could be confirmed that neuronal activation patterns in the ventromedial prefrontal cortex during extinction can predict the retrieval of extinction memory 24 hours later. In contrast, an effect of levodopa on extinction memory has not been confirmed.

In general, this is a very elaborate and in large parts very well controlled study with sufficient statistical power. Replication study guidelines were followed, and all unregistered analyses were clearly labelled. The results of the replication study are of great interest to the field. In addition, the relationship between salivary alpha-amylase activity and locus coeruleus connectivity and its importance for extinction retrieval is investigated. The results show that levodopa can enhance extinction consolidation during high arousal. The relevance of the results is discussed for exposure therapy sessions, which are typically associated with high arousal.

The latter results in particular are extremely interesting and may make an important contribution to our understanding of extinction retrieval.

However, from a methodological perspective, they are very difficult to assess. In particular, the manuscript has weaknesses in the sampling and analysis of salivary alpha-amylase and in the measurement of locus coeruleus connectivity patterns.

1) Salivary alpha-amylase (sAA).

The key to reliable interpretation of salivary alpha-amylase activity is sampling and analysis. Both are not well described in the manuscript, and some standards may not have been followed. Therefore, it is important that essential information is added first in order to interpret the data appropriately.

- It is unclear whether stimulation of sAA occurred. For example, chewing can stimulate sAA secretion. Accordingly, it is important to know the exact instructions that subjects received during saliva collection. Was there any standardization of saliva collection? How long were the collectors kept in the mouth? Were subjects instructed to chew on the collectors or did they simply place them in the mouth? Also, since secretion of sAA varies from salivary gland to salivary gland, it is important to know where the collector was placed when it was not moved in the mouth. Please fill in the information as accurately as possible.

- Cotton swabs are rather unsuitable for SAA analyses. They cannot be completely centrifuged and the amylase starts to decompose the absorbent cotton, i.e. the enzyme activity increases (therefore synthetic collectors are recommended). In any case, you should specify exactly which collectors were used (e.g. "Sallivettes white cap" with cotton swabs, "Manufacturer") and how much time elapsed between sample collection and freezing and thawing and analysis. Please fill in the information as accurately as possible.

- To my knowledge, the sAA tests do not determine the sAA concentration in saliva, but the metabolic activity in units per ml. SAA concentration is a combined effect of flow rate and protein secretion. This should be corrected.

AUTHORS: Firstly, allow us to reiterate our introductory remarks to this point-by-point reply that the manuscript has undergone significant changes in response to the reviewer comments, particularly with the inclusion of the new finding that L-DOPA effectiveness in enhancing extinction memory consolidation is predicted by trait sAA (inspired by reviewer 2). Indirectly, this finding will probably alleviate already some of this reviewer's concerns, especially due to the very reliable nature of the baseline measures of sAA activity on the three experimental days, which served to determine trait sAA in our new analyses. (Please see our first reply to reviewer 2 above, the detailed report of these findings in the Results, starting L275, and our literature review on trait-like effects in sAA on L159 and L453 ff., which also revealed substantial heritability of this measure and the possibility to use average sAA measures in a state of rest, or baseline, from as little as three days to obtain reliable trait markers.) Comparatively, the acute reactions of sAA to the experimental manipulation on day 3 ('state sAA' in the further), potentially indicating an arousal response, appear to be of lesser importance for extinction learning and memory and for the L-DOPA effect.

We would also like to apologize right away that we spoke of alpha-amylase concentrations in the previous version of the manuscript. This has now been corrected.

With regards to the methodology used for sAA sample collection, we have now added the additional information in the method section (L717 ff.) and also discuss its limitations, starting L518. In brief, saliva sample collection followed a standardized procedure using cotton swabs from the Sarstedt manufacturer. Participants were instructed to place the swab in their cheek without chewing on it. The swab remained in the cheek timed for one minute to allow for sufficient saliva collection. Samples were collected throughout each day's experiment, and they were stored in the fridge approximately 10-20 minutes after the experiment's completion. The elapsed time from the first sample collection (baseline) to freezing was approximately 100 minutes for day 1, 160 minutes for day 2, and 70 minutes for day 3. All samples remained frozen throughout the data collection period

of three years, and they were thawed one day prior to sending them to the analyzing laboratory. All samples were analyzed within two weeks after being sent to the laboratory.

To assess stability and test-retest reliability of the baseline measures that served to establish trait sAA, we calculated their intercorrelations between the three days (all highly significant, see L282)) and the intraclass correlation coefficient (ICC), which indicated high stability (ICC = 0.701, $p < 0.0001$).

We acknowledge the limitation of using cotton swabs for sAA collection and will strive to use better methodology in a further planned replication study.

REVIEWER: - Finally, the direct relationship between sAA and noradrenergic/sympathetic activity is not well established. Reactivity measures such as the increase in sAA are useful for a cautious estimate of noradrenergic or sympathetic activity (e.g., Dietzen et al., 2014; <http://dx.doi.org/10.1016/j.biopsycho.2014.08.001>). However, in the present study, no reactivity measures were used, but the sAA values before extinction. These values are very difficult to interpret, which is a clear limitation.

- The Interpretation of amylase activity as a direct measure of physiological excitation is also difficult unless it is caused by a unique condition or can be confirmed by behavioural data.

The following paper provides a good overview of the points mentioned above:

- Reference: Bosch, J. A., Veerman, E. C. I., de Geus, E. J., & Proctor, G. B. (2011). α -Amylase as a reliable and convenient measure of sympathetic activity: Don't start salivating just yet! *Psychoneuroendocrinology*, 36(4), 449–453. <https://doi.org/10.1016/j.psyneuen.2010.12.019>

AUTHORS: We agree with the reviewer that the direct relationship between sAA reactivity and noradrenergic/sympathetic response is not well established and that changes in sAA activity in response to a stimulus are not a very reliable marker of stimulus-induced increases in noradrenergic or SNS activity. See also L451. We also acknowledge that our study did not include specific reactivity measures, simply because at the group level there was no detectable reaction relative to baseline on the experimental day 2 to participants being placed in the scanner and receiving extinction training. This is now described in more detail L267 ff. and discussed L445 ff.

Our new analyses therefore focused on trait sAA, see above. In order to also take into account potential influences of an acute state of arousal or SNS activity around the time of extinction in our participants, we therefore resorted to calculating the difference between an individual's peak sAA activity level on day 2 (observed either at the baseline measure immediately before the experiment or at the time point just after extinction) from their trait sAA value and label this 'state sAA'. This measure is non-collinear to trait sAA and could thus be included with trait sAA in the same model, which we used to predict either extinction retrieval (L296 ff.) or extinction learning (L315 ff.) or vmPFC reactivations (L331 ff.). These analyses revealed some influences of state sAA, in particular on vmPFC reactivations.

REVIEWER: 2) Use of the locus coeruleus as a seed region for connectivity measurements

- The LC is very small and surrounded by large blood vessels and ventricles. Functional measurements of the structure have high requirements for adequate mapping of the structure. A 2009 paper by Keren is cited as a reference for measuring the LC. However, a debate arose about the reliable measurement of the activity of the LC and similar small structures. It has been suggested that many results may be erroneous because mapping of the LC is very difficult, and pulsation of nearby blood vessels and motion artifacts due to proximity to the ventricles can greatly affect measurements.

Reference: Astafiev, S. V., Snyder, A. Z., Shulman, G. L., & Corbetta, M. (2010). Comment on "Modafinil Shifts Human Locus Coeruleus to Low-Tonic, High-Phasic Activity During Functional MRI" and "Homeostatic Sleep Pressure and Responses to Sustained Attention in the Suprachiasmatic Area". *Science*, 328(5976), 309–309.

- In an eLetter published along with the publication, <https://www.science.org/doi/10.1126/science.1177200>

the editors and other experts in the field ultimately recommend that future studies ...

- Perform explicit brainstem normalization
- Use standardized LC masks as ROIs
- And should perform cardiac gating to control pulsation

I am aware that the above points are not fully feasible in such an elaborate study. However, it would increase confidence in the data if the connectivity maps were shown within the LC masks. For this purpose, the regions whose LC connectivity directly correlates with sAA (vSTR_L, vSTR_R, dmPFC) could be used as seed regions in control analyses. Subsequently, it can be visualized whether their connectivity patterns in the brainstem lie within standardized masks of the LC. If not already done, brainstem normalization would be required.

Example of LC masks related to the above eLetter could be found here: www.eckertlab.org/lc

Please also indicate if the left and right LC time series were averaged for the analysis in the initial analysis or if the left and right LC time series were specified individually in the model.

AUTHORS: Before we address this important point, we would like to summarize some additional analyses we did prior to changing the preprocessing of the rsfMRI data used for LC connectivity, as requested by reviewer 3. With reference to the last comment of reviewer 2, we found that LC connectivity at baseline, determined with the old method, cannot predict corresponding sAA at day 1 and 3. An additional analysis aimed at predicting mean baseline sAA (averaged over days 1-3) was also not successful. Unlike baseline sAA values, which were characterized by a high reliability score (ICC=0.7, see above), functional connections between the LC and other brain regions of the LC network showed no or low reliability (ICC=0-0.3). Together, these results strongly dampened our initial enthusiasm and raised concerns about a true link of sAA and LC connectivity.

In our previous approach, no brainstem-specific normalization was performed and rsfMRI data were preprocessed using the software package fMRIPrep, which we thought was appropriate, as a recent study on different techniques to obtain LC-FC networks showed a remarkable overlap between FC maps from whole-brain normalized data combined with a single MNI-space LC mask and FC maps calculated using individual seed-regions obtained from neuromelanin-sensitive scans (Mäki-Marttunen, 2020, <https://doi.org/10.1016/j.neuroimage.2020.117409>). However, in agreement with reviewer 3, we see the risk for inaccurate normalization in our previous analysis and revised the preprocessing procedure. We therefore used SPM instead of fMRIPrep, as brainstem-normalization is not implemented in fMRIPrep.

rsfMRI data were first preprocessed as described in the task-fMRI methods section (L752) with the only difference that a 3mm smoothing kernel was applied. For LC signal extraction, rsfMRI data obtained after realignment were furthermore normalized to the MNI-based cerebellum-brainstem template using the SPM toolbox SUIT (Diedrichsen, 2006; [10.1016/j.neuroimage.2006.05.056](https://doi.org/10.1016/j.neuroimage.2006.05.056)). The procedure involved the following steps: 1) coregistration of the anatomical image and the realigned fMRI images, 2) isolation of the brainstem and cerebellum from the anatomical image, 2) further manual cropping of the isolated brainstem/cerebellum mask, 3) normalization with a nonlinear deformation map to the brainstem template and application of the calculated deformation field to the functional images.

LC networks were calculated from baseline rsfMRI data for days 1, 2 and 3 as described in the previous version of the manuscript using a standardized LC mask. Additionally, we calculated a single LC-FC network from the concatenated rsfMRI baseline scans of day 1-3. This LC-FC network was used to predict the averaged baseline sAA. We also calculated an ICC score for each region of the LC-FC network indicating the reliability of the region's FC to the LC across days 1, 2 and 3.

Brainstem-specific normalization yielded LC-FC networks that differed markedly in architecture from those computed by fMRIPrep (e.g., striatal regions appeared only in the fMRIPrep variant), which raises concerns about the general robustness of LC network calculations. As suggested by the

reviewer, a final assessment of whether brainstem normalization yields better results than the old method could be achieved by a control analysis in which brain regions showing positive FC with the LC are used as seeds to determine whether there is specific connectivity with voxels within the LC mask. We refrained from such an analysis because, similar to the previous approach, the FC networks resulting from the new procedure did neither predict day-specific sAA nor the averaged (trait-like) sAA. As for fMRIPrep-preprocessed data, LC connections were characterized by very low ICC scores between 0 and 0.3. We therefore decided to remove the LC-FC analysis from the revised version of the manuscript.

We thank the reviewer for their important remarks that have protected us against publishing a false positive finding.

REVIEWER: Other points:

3) Figure 7.

If I understand correctly, ROI-to-ROI whole-brain analyses were performed over 400 cortical and subcortical regions, not seed-to-voxel analyses. If the ventral striatum was one of the ROIs, why is there a light blue cluster in the left posterior putamen in Figure 7? Please explain and describe the results in more detail.

What is the difference between dACC 1 and dACC2? Please explain.

AUTHORS: After careful consideration, we have decided to omit the previous analyses due to the challenges associated to it.

REVIEWER: 4) Replication:

- Please briefly state how the technical parameters / scanner / electroshock ect. were comparable to the initial study.

AUTHORS: All technical parameters, including the MRI scanner and equipment, were kept identical to those used in the initial study. See L547.

REVIEWER: 5) Statistical trends:

- Trend wise association are non-significant results and should be presented as such.

AUTHORS: This is now done throughout the manuscript.

REVIEWER: 6) Stress

-In the study, stress was not explicitly measured (e.g., via a questionnaire). Levels of sAA and cortisol varied depending on many factors and could not be interpreted as independent markers of stress. No measures of sAA reactivity were reported, and it was not tested whether there was an increase in cortisol above the normal pulse rate (responder analysis). This is also not possible with the two saliva samples before and after the respective measurements. I therefore advise against using the term stress and arguing with more extensive stress regulation patterns. Statements like:
„ sAA and sCORT originate from two different stress systems, the first being faster and more sensitive in responding to stressors, while the second responds more slowly and to more stressful tasks. Our current results suggest that acute arousal system activation (related to sAA) might have stronger direct effects on extinction than HPA system activation (related to sCORT) and show more important interactions with dopaminergic transmission.“

suggest that there was a solid activation of both stress systems. However, this was not tested and so it is unclear whether the sAA and cort values reflect a stress response or the normal circadian course.

AUTHORS: The reviewer is right that we cannot demonstrate solid activation of the two stress systems and all corresponding claims have been dropped.

REVIEWER: The journal explicitly asks the reviewers to pay attention to compliance with the SAGER guidelines and to check whether the analyses are in accordance with the preregistration.

- In the present study, only men were examined, which is methodologically justified.
- However, the SAGER guidelines require that it should be made clear in the title and abstract that only one sex is involved. The title and abstract should be adjusted accordingly.

- The discussion should include a point in which it is made clear that the results apply to only one sex.

- Based on my review, the analyses do not differ from the preregistered analyses. Some analyses have been omitted due to technical issues. Additional analyses have been clearly identified.

AUTHORS: The author is right that in the case research findings apply to only one sex or gender, that it must be indicated in the title and/or abstract. We have now added this information in the abstract in L15 as well as in the discussion in L355 and in L531 ff.

REVIEWERS' COMMENTS

Reviewer #1 (Remarks to the Author):

The authors have addressed all of my comments.

Reviewer #2 (Remarks to the Author):

The authors have done a commendable job addressing prior reviewer concerns and comments. My only remaining comment is that more attention should be directed to the observation that sAA was not correlated with state or trait anxiety. This is relevant for clearly dissociating the larger construct of "arousal" with what the authors are measuring here with specific marker of sAA. sAA, and other physiological measures, may be one component, but not the only component of "arousal". For example, one can be emotionally aroused without corresponding physiology (see the literature on desynchrony of measures of emotion).

Reviewer #3 (Remarks to the Author):

The authors have put a lot of effort into revising the manuscript. The critical analyses on LC activity were omitted. Missing information on methodology has been added. I have no further concerns. Good luck with this exciting manuscript.

Point-by-point reply

“Trait salivary alpha-amylase activity levels in males define the conditions for facilitation by L-DOPA of extinction consolidation”

Andres et al.

Introductory remarks to all reviewers

We would like to express our gratitude for the valuable feedback and comments on our manuscript. We are pleased to see that the majority of the reviewers’ concerns have been addressed to their satisfaction, and we appreciate their time and effort in reviewing our work.

REVIEWERS' COMMENTS

Reviewer #1 (Remarks to the Author):

REVIEWER: The authors have addressed all of my comments.

Reviewer #2 (Remarks to the Author):

REVIEWER: The authors have done a commendable job addressing prior reviewer concerns and comments.

My only remaining comment is that more attention should be directed to the observation that sAA was not correlated with state or trait anxiety. This is relevant for clearly dissociating the larger construct of "arousal" with what the authors are measuring here with a specific marker of sAA. sAA, and other physiological measures, may be one component, but not the only component of "arousal". For example, one can be emotionally aroused without corresponding physiology (see the literature on desynchrony of measures of emotion).

AUTHORS: We appreciate the reviewers’ comment and have now included the following sentences in the discussion: It is also worth mentioning that self-reported trait and state anxiety did not correlate with trait or state sAA in our data, highlighting that trait sAA represents just one facet of emotional arousal. The concept of arousal is multifaceted, encompassing subjective, behavioral, and physiological components, and response systems can be desynchronized (Hodgson & Rachman, 1974; Stam & McEwan, 1987). Our results suggest that trait sAA, rather than self-reported anxiety, is the relevant factor for the facilitation of consolidation with L-DOPA. (L525 ff.)

Reviewer #3 (Remarks to the Author):

REVIEWER: The authors have put a lot of effort into revising the manuscript. The critical analyses on LC activity were omitted. Missing information on methodology has been added. I have no further concerns. Good luck with this exciting manuscript.